# Analysis of Zika virus capsid-*Aedes aegypti* mosquito interactome reveals pro-viral host factors critical for establishing infection

Rommel J. Gestuveo [1,2✉], Jamie Royle [1], Claire L. Donald [1,5], Douglas J. Lamont[3], Edward C. Hutchinson [1], Andres Merits[4], Alain Kohl [1✉] & Margus Varjak [1,4✉]

The escalating global prevalence of arboviral diseases emphasizes the need to improve our understanding of their biology. Research in this area has been hindered by the lack of molecular tools for studying virus-mosquito interactions. Here, we develop an *Aedes aegypti* cell line which stably expresses Zika virus (ZIKV) capsid proteins in order to study virus-vector protein-protein interactions through quantitative label-free proteomics. We identify 157 interactors and show that eight have potentially pro-viral activity during ZIKV infection in mosquito cells. Notably, silencing of transitional endoplasmic reticulum protein TER94 prevents ZIKV capsid degradation and significantly reduces viral replication. Similar results are observed if the TER94 ortholog (VCP) functioning is blocked with inhibitors in human cells. In addition, we show that an E3 ubiquitin-protein ligase, UBR5, mediates the interaction between TER94 and ZIKV capsid. Our study demonstrates a pro-viral function for TER94/ VCP during ZIKV infection that is conserved between human and mosquito cells.

[1] MRC-University of Glasgow Centre for Virus Research, Glasgow, UK. [2] Division of Biological Sciences, University of the Philippines Visayas, Miagao, Iloilo, Philippines. [3] FingerPrints Proteomics Facility, School of Life Sciences, University of Dundee, Dundee, UK. [4] Institute of Technology, University of Tartu, Tartu, Estonia. [5] Present address: Institute of Molecular, Cell and Systems Biology, University of Glasgow, Scotland, UK. ✉email: rjgestuveo@up.edu.ph; alain.kohl@glasgow.ac.uk; margus.varjak@glasgow.ac.uk

Zika virus (ZIKV) is an arbovirus of the *Flaviviridae* family[1–4] originally isolated from a Rhesus monkey in Uganda in 1947[5,6]. Though previously limited to Africa and Southeast Asia, outbreaks across Pacific Ocean islands were detected throughout 2013–2014[7,8]. During the recent outbreak in South and Central America in 2015–2016, the yellow fever mosquito (*Aedes aegypti*) was identified as the key ZIKV vector[9–12]. ZIKV infection is associated with mild and unspecific disease manifestations but has also been linked to serious complications including Guillain-Barré syndrome and congenital Zika syndrome[13–19]. Currently no drugs or vaccines against ZIKV are available, which enforces the need for complementary vector research in mosquito cells to better understand its replication, transmission, and interaction with cellular proteins[20,21].

The ZIKV RNA genome is translated into a single polyprotein processed by host and viral proteases into structural and non-structural proteins[22–24]. ZIKV enters cells via receptor-mediated endocytosis[25] and is trafficked in endosomes where lowering the pH induces membrane fusion[26], exposing the nucleocapsid to the cytoplasm for uncoating. However, nucleocapsid disassembly is less well understood[27,28]. There are two forms of ZIKV capsid proteins in infected cells (Fig. 1a). Initially, anchored capsid (AC; 122 amino acids) is bound to the endoplasmic reticulum (ER) membrane by its trans-membrane helices. During viral replication, AC is cleaved by viral proteases, releasing an untethered capsid (C; 104 amino acids) that assembles with the RNA genome for release[23,29,30]. Despite its small size, protein–protein interaction (PPI) studies have shown that ZIKV capsid interacts with several human cellular pathways and affects intracellular functions during viral replication[31,32]. These PPI studies in mammalian systems[33–42] have been pivotal in understanding ZIKV biology[43]. However, PPIs are largely unexplored in mosquito cells due to the lack of molecular tools required for such studies.

Here, we identify protein interactors of ZIKV anchored and untethered capsid in *Ae. aegypti* cells by developing AF5 stable cell lines expressing ZIKV capsid proteins, allowing mass spectrometry-based experiments to be carried out. We identified 157 potential interactors, with 38 shared between C and AC. A knockdown screen of 24 shared interactors showed that eight were pro-viral host factors. This set of pro-viral interactors included transitional endoplasmic reticulum 94 (TER94), whose human ortholog is valosin-containing protein (VCP/p97). TER94 functions with a number of co-factors to segregate target proteins from protein assemblies following the ubiquitin-proteasome pathway (UPP)[44–47]. Our results suggest that TER94 targets ZIKV capsid for proteasomal degradation in AF5 cells. We found that this role is conserved between mosquito and human cells and that TER94/VCP is needed during the early stages of infection. We also identified a potential co-factor, ubiquitin protein ligase E3 component N-recognin 5 (UBR5), involved in mediating TER94/VCP-ZIKV capsid interactions in mosquito and human cells. These results suggest that TER94/VCP plays an important role in viral RNA uncoating by interacting with ZIKV capsid and subsequently trafficking it for proteasomal degradation. This study increases our understanding of ZIKV capsid functions and opens a door to pursue further PPI studies in mosquito cells. These studies will be required to understand the commonalities and differences between arboviral interactions with host proteins in vector and mammalian cells.

## Results

**Developing stable mosquito cell lines to study ZIKV capsid-*Ae. aegypti* PPI.** To study the ZIKV capsid interactome in mosquito cells, AF5 cells stably expressing ZIKV C or AC proteins were developed. Coding sequences for C or AC from a ZIKV infectious cDNA (icDNA) clone, pCCI-SP6-ZIKV[48] were cloned into an expression plasmid under the control of an *Ae. aegypti* poly-ubiquitin (PUb) promoter[49], with a Zeocin resistance gene (ZeoR) separated from V5-tagged C or AC by two copies of 2A autoprotease (Fig. 1a). Plasmids were linearized and transfected into AF5 cells followed by ZeoR selection to obtain stable cell lines. Immunostaining of AF5-V5-AC and AF5-V5-C cells revealed capsid localization along the perinuclear region, with V5-C appearing to be more dispersed, while minimal speckling of V5-AC was observed (Supplementary Fig. 1a).

The pipeline for studying ZIKV capsid interactome in stable mosquito cell lines is straightforward and can be easily adapted to other viral proteins (Fig. 1b). This system involves immunoprecipitation (IP) of V5-C, V5-AC, and V5-eGFP (as a control for background binding) using an anti-V5 tag antibody (Supplementary Fig. 1b). Three independent pulldown samples from each cell line were subjected to nano-liquid chromatography and tandem mass spectrometry (nLC-MS/MS) under label-free (LFQ) conditions. Potential interactors were selected based on (1) consistent identification with at least two peptide spectral matches in all V5-C or V5-AC biological replicate samples and absent in the V5-eGFP controls, (2) intensity-based absolute quantification (iBAQ) intensity normalization[50] ranking, and (3) SAINTq[51] analysis (Supplementary Fig. 1c).

To construct PPI networks, UniProt IDs were converted to *Ae. aegypti* gene stable IDs in VectorBase[52] (AaegL5.3 release) and interaction networks generated in StringDB[53] with a cut-off combined score of at least 0.7. Networks were visualized in Cytoscape[54] and gene ontology (GO) and pathway enrichment analyses performed in VectorBase and DAVID[55]. Orthologs of interactors in *Drosophila melanogaster* or *Homo sapiens* were confirmed through OrthoDB[56] or by conducting homology searches using BLASTP algorithm. These were mapped using FlyBase[57] (FB2020_05 release) and HGNC[58], respectively. Inferred annotations were used to characterize the generated networks with data mining for potential PPI in BioGRID[59].

The selection criteria of the interactome data resulted in the stringent identification of 148 and 47 potential mosquito protein interactors with ZIKV C and AC, respectively (Venn diagram, Fig. 1b; Supplementary Tables 1 and 2), and 38 with both C and AC. A number of ribosomal proteins were identified, while other enriched pathways clustered within the network contained several structural and nucleotide-binding proteins and transporters (Fig. 1c). The relative abundance of proteins in the interactomes allowed us to estimate their relative frequency of binding to the bait proteins. For both C and AC interactors, AAEL015065 (*Ae. aegypti* SPTAN1) was the most abundant interacting protein based on the number of peptide spectral matches. AAEL003530 (*Ae. aegypti* RPLP1) had the highest relative abundance using the more sophisticated iBAQ and LFQ quantitation algorithms, which normalize peptide spectral intensity by the predicted peptide yield of each protein. GO analysis (Fig. 2a) showed that most of the interactors were involved in translation (GO:0006412), metabolic processes (GO:0003461 and GO:0009058), or as structural components of ribosomal complexes (GO:0005840 and GO:0003735). Notably, ZIKV capsid also interacted with pathways involved in carbon fixation, tryptophan and fatty acid metabolism, and with a number of proteasome subunits.

Interestingly, capsid was shown to interact with proteins of the RNA interference (RNAi) pathway including Dicer-2 (Dcr2; AAEL006794). The RNAi pathway, in particular the exogenous siRNA pathway in which Dcr2 plays a key effector role, is an antiviral response found in mosquitoes and other insects[60–63]. Previously, it was shown that yellow fever virus (YFV) C protein

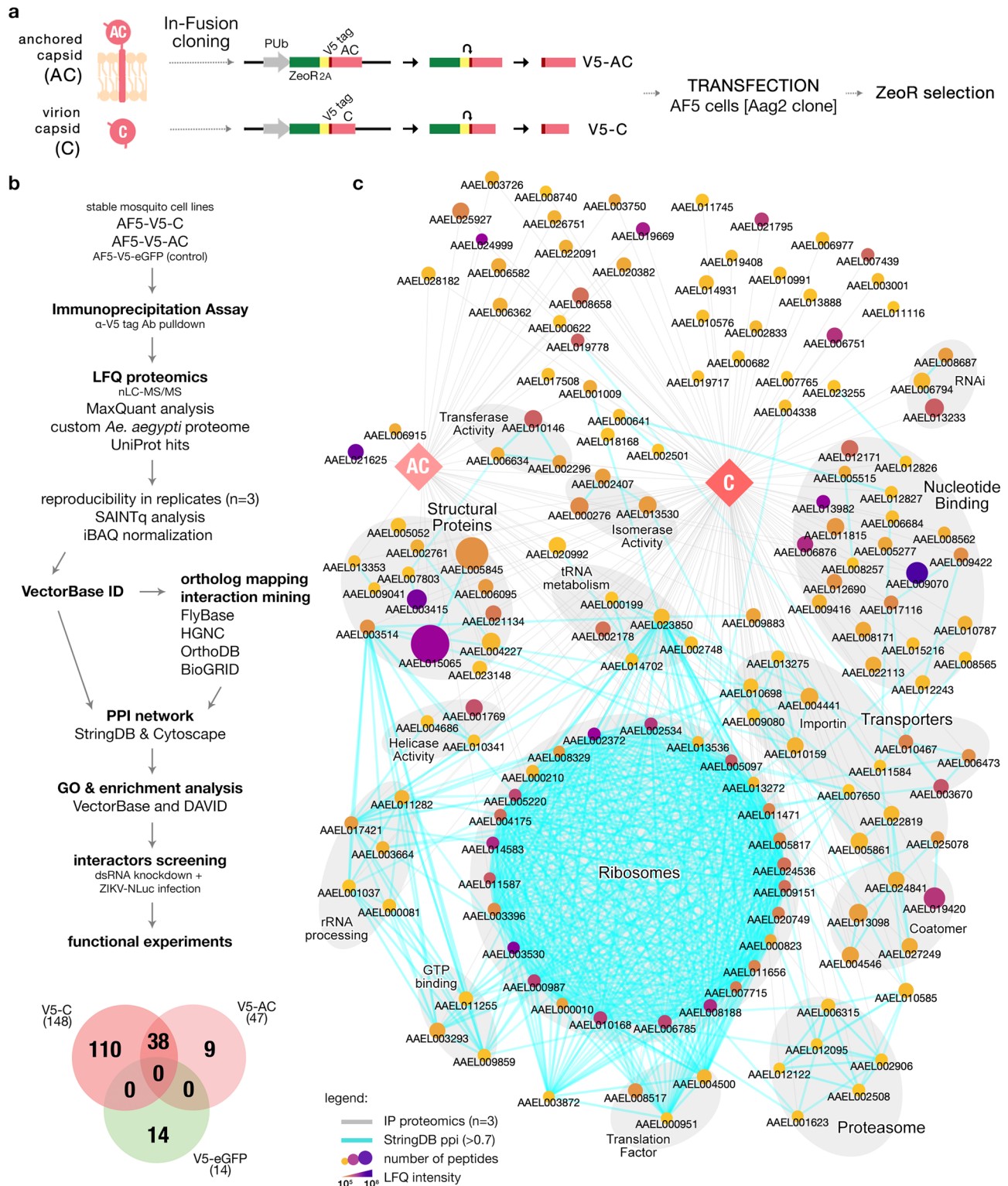

**Fig. 1 Developing *Ae. aegypti* Stable Mosquito Cell Lines for Studying ZIKV C and AC PPI. a** Schematic showing plasmid constructs used to express V5-tagged ZIKV C or AC under the control of a PUb promoter with ZeoR gene, and duplicated 2A autocleavage sequence. **b** Study pipeline using developed *Ae. aegypti*-derived AF5 cell line stably expressing ZIKV C or AC to investigate vector-virus PPIs. Venn diagram showing the distribution of potential interactors among immunoprecipitated proteins. **c** ZIKV capsid-*Ae. aegypti* PPI network of 157 interactors from IP proteomics. C and AC nodes (diamonds) with IP proteomics interaction (gray edges) to host interactors (circles) and StringDB PPI (cyan edges). Gray highlights enclose nodes with broadly related functions. List of protein interactors provided in Supplementary Tables 1 and 2.

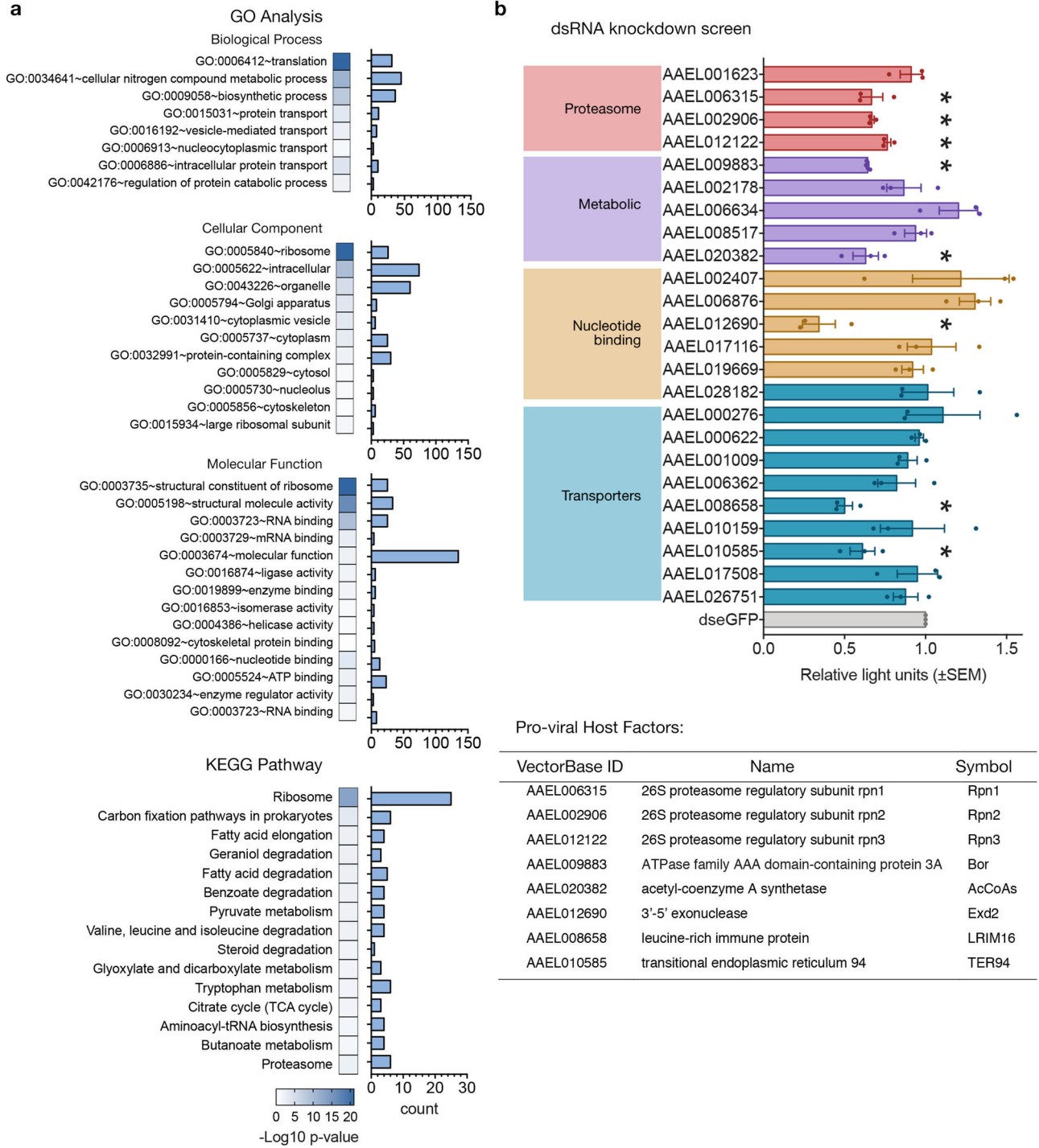

**Fig. 2 ZIKV capsid interactors and dsRNA knockdown screen. a** GO and KEGG pathway enrichment analyses of annotated interactors performed in VectorBase and DAVID. Heat map shows significant enrichments (*p*-value < 0.05 in −log10) with Benjamini correction against the background gene set with actual *p*-values provided in the Source Data file. Bars indicate gene count. **b** dsRNA knockdown screen of 24 interactors (Supplementary Table 3) in AF5 cells with ZIKV-NLuc (MOI = 1) infection 24 hpt for 72 h. NLuc levels presented as mean ± SEM light units relative to dseGFP controls set to 1 from *n* = 3 independent repeats. *$p$-value < 0.05 determined by two-tailed Student's *t*-test, where vs. DMSO controls: AAEL006315 $p = 0.041$; AAEL002906 $p = 0.002$; AAEL012122 $p = 0.008$; AAEL009883 $p = 0.001$; AAEL020382 $p = 0.042$; AAEL012690 $p = 0.023$; AAEL008658 $p = 0.009$; AAEL010585 $p = 0.037$; other *p*-values provided in the Source Data file.

is a suppressor of RNAi, and the same effect has been proposed for ZIKV capsid[64]. However in a separate study, ZIKV AC expression was found to benefit an unrelated alphavirus, Semliki Forest virus (SFV) in a Dcr2-independent manner[65]. To assess this, RNAi sensor assays[66] were performed in the stable cell lines by co-transfecting firefly luciferase (FFLuc) expression plasmid (pIZ-Fluc) with dsRNA or siRNA targeting FFLuc. In this assay, expression of ZIKV C or AC in the stable cell lines did not affect mosquito RNAi activity, regardless of whether dsRNA or siRNA was used to trigger silencing activity. This suggests that ZIKV C and AC do not suppress the RNAi pathway in these cells, at least (Supplementary Fig. 1d).

Comparing the human orthologs of our mosquito interactors with previous PPI studies[35–37], interactors with ZIKV capsid in human cells showed minimal overlap between proteomics data sets and only one protein, GNL2, was common to all data sets (Supplementary Fig. 1e, left panel). In addition, when comparing our interactors to previous CRISPR genome-wide screens in human cells[67–69] during ZIKV infection, no common host factors were identified (Supplementary Fig. 1e, right panel). The very little overlap between our interactors with previous studies highlights the cell-type and technique-specific effects of performing such studies to investigate host/vector-virus interactions.

To test some of these interactions, a dsRNA knockdown screen was performed on 24 of the 38 shared potential interactors of C and AC (Fig. 2b, Supplementary Table 3). AF5 cells were transfected with gene-specific dsRNAs and infected 24 h post-transfection (hpt) with ZIKV-NLuc[48], a reporter virus expressing Nanoluciferase. Results analyzed 72 h post-infection (hpi) showed out of the 24 potential interactors tested, eight were identified to have pro-viral activity during ZIKV infection (Fig. 2b, inserted table). Interestingly, all pro-viral proteasome subunits: AAEL06315 (Rpn1), AAEL002906 (Rpn2), and AAEL012122 (Rpn3), cluster together with TER94 (AAEL010585) in a subnetwork involving the proteasome (Fig. 1c).

**ZIKV C interacts with TER94-UPP and is important during virus replication in mosquito cells**. As part of the UPP that is responsible for the majority of protein degradation in cells[70,71], TER94/VCP acts as a chaperone[45,72,73] for ubiquitinated proteins but can also perform other functions through co-factors that facilitate binding with target proteins[47,74]. To determine the effect of silencing TER94 or the proteasome subunit Rpn1 (as a representative) on V5-C stability, knockdown experiments with a cycloheximide chase assay were performed in AF5-V5-C cells (Fig. 3a). The abundance of V5-C was more stable over time when TER94 and Rpn1 were silenced, compared to the dseGFP-transfected control (Fig. 3b). This indicates that ZIKV capsid may undergo proteasomal degradation through ubiquitination, similar to what has been shown for dengue virus (DENV) C protein[75].

The interaction of ZIKV C and TER94 was confirmed by co-immunoprecipitation (co-IP) assays using AF5-V5-C stable cell lines transfected with a PUb expression plasmid[66] encoding a myc-tagged TER94. Since TER94 is involved in the UPP leading to target protein degradation, the co-IP was carried out in the presence of the proteasome and deubiquitinase (DUB) inhibitors, MG132 and ML364, respectively. IP of myc-TER94 with anti-myc antibody resulted in the pulldown of V5-C, while reciprocal assays revealed pulldown of myc-TER94 by anti-V5 antibody (Fig. 3c), thus confirming the interaction.

The pro-viral role of TER94 and Rpn1 was investigated further by performing dsRNA knockdowns (silencing did not affect cell viability; Supplementary Fig. 1f), followed by infection at 24 hpt with ZIKV PE243[76], a patient-derived ZIKV isolate. TER94 and Rpn1 silencing significantly reduced ZIKV RNA levels as well as virus titers when compared to the dseGFP controls (Fig. 3d). The reduced levels of ZIKV under knockdown conditions may have affected the viral replication cycle either early during genome replication or later in virion assembly, which has been shown to be UPP-dependent in DENV[77].

To investigate whether TER94 and Rpn1 are required during ZIKV replication, a ZIKV derived replicon system was used (Fig. 3e, top panel). This allows the investigation of genome replication without active infection and bypasses virus entry and assembly steps without producing infectious virus particles, unlike ZIKV-NLuc virus. AF5 cells under TER94 and Rpn1 knockdown conditions were co-transfected with in vitro transcribed ZIKV NLuc-expressing replicon RNA, abbreviated from here on to ZIKV Replicon for simplicity (Fig. 3e, bottom panel). Luciferase levels measured 24 hpt showed that TER94 and Rpn1 knockdown did not significantly affect ZIKV Replicon activity relative to the dseGFP control (Fig. 3f). This suggests that genome replication is not directly affected by the knockdown of TER94 or Rpn1.

Since TER94 and Rpn1 act primarily on ubiquitinated proteins, this led us to hypothesize that ZIKV C may also be ubiquitinated, in a similar way to DENV nucleocapsid at the start of infection[75]. To test this, an assay to detect ZIKV genome at early stages of infection was designed (Fig. 3g). AF5 cells were transfected with dsTER94 or dsRpn1 to knockdown host genes; dseGFP was used as a non-targeting control. At 24 hpt, ZIKV PE243 was applied to the cells and allowed to bind for 30 min on ice. The cells were returned to 37 °C for simultaneous entry of virus particles. The inoculum was aspirated to remove any unadhered virus particles and replaced with fresh medium containing cycloheximide to stall ribosomes on viral RNA. This prevented the synthesis of viral proteins and thereby the generation of new viral RNAs.

At 1 and 2 hpi, internalized viral genomes were quantified by RT-qPCR. Results showed that dsTER94 transfected cells had significantly higher detectable ZIKV RNA compared to the dseGFP control at 1 hpi but this was not the case at 2 hpi (Fig. 3h). On the other hand, Rpn1 silenced cells showed no difference in detectable ZIKV RNA at either timepoints compared to controls. This suggests that proteasomal degradation at early timepoints during infection may not be necessary to release the viral genome and may be a by-product of C interaction with TER94. The importance of capsid ubiquitination at early stages of infection has been previously shown for DENV[75]. However, it is likely that the knockdown of TER94 inhibited the release of ubiquitinated capsid bound to the RNA genome, preventing uncoating. This protected and preserved the genome, limiting degradation and resulting in more detectable viral RNA at 1 hpi. It has been shown that as flavivirus RNA genome is released into the cytoplasm, it is degraded rapidly[75,78] as exposed viral RNA becomes accessible to nucleases[79].

**Ortholog mapping of mosquito interactors reveals human TER94 is vital during ZIKV infection**. TER94 is a highly conserved protein across invertebrate and vertebrate species. It is widely studied owing to its physiological importance of maintaining protein homeostasis in cells[80,81]. In *D. melanogaster*, dysregulation of this protein has been linked to retinal degeneration in adult flies and developmental issues of larvae[82,83], while in humans mutations have been implicated in several proteinopathies and malignancies[44,45,84]. VCP has also been shown to be important in viral infections of avian coronavirus[85], human cytomegalovirus[86], Sindbis virus (SINV)[87], chikungunya virus[88], West Nile virus[89], hepatitis C virus[90], and more recently YFV[91]. Its prominent role in the UPP and viral infections merits additional investigation during ZIKV infection in human cells.

Ortholog mapping across VectorBase, FlyBase, and HGNC was performed for all mosquito protein interactors of ZIKV C and AC associated with TER94 and proteasome subunits. These proteins and their PPIs were highly conserved across *D. melanogaster* and *H. sapiens* (Fig. 4a). To test the role of VCP during ZIKV infection in human cells, two siRNAs targeting VCP (siVCP-1, siVCP-2, or the combination of both) were transfected into A549 cells followed by infection with ZIKV-NLuc or ZIKV PE243. VCP knockdown significantly reduced both reporter virus readings and viral RNA compared to controls (Fig. 4b). Knockdown efficiency of siRNAs was confirmed by immunoblots with no effect on cell viabilities (Supplementary Fig. 2a, b).

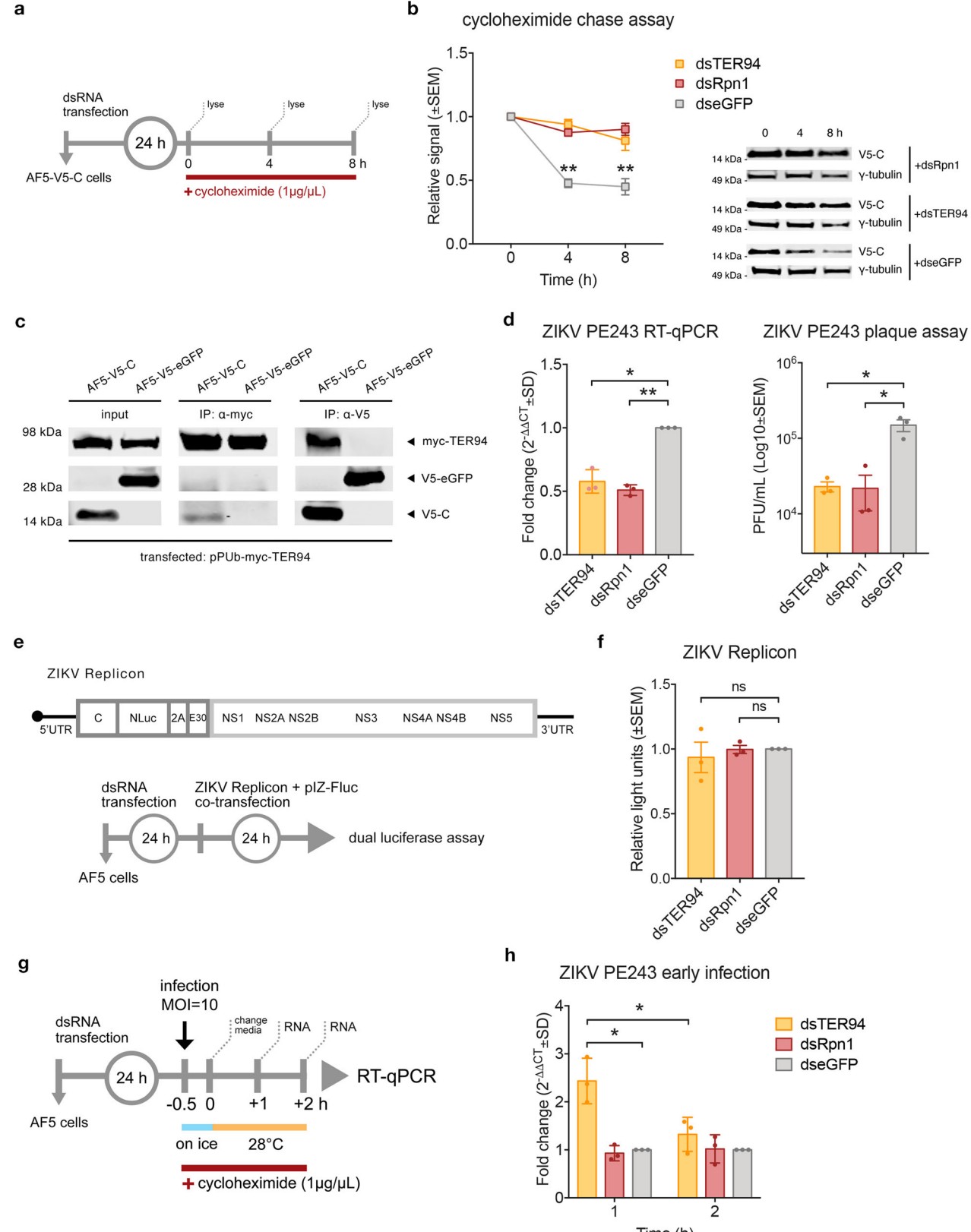

Verified and readily available chemical inhibitors for human UPP proteins were used to inhibit specific proteins or steps in the pathway (Fig. 4c) to determine those important during ZIKV replication. Pyr-41 was used to inhibit ubiquitin recruitment by E1 ligase preventing ubiquitination; VCP function was blocked by DBeQ and EerI; DUB activity was suppressed by ML364; and

proteasomal degradation was avoided by treating with MG132[75,92–94]. Cell viability assays were performed prior to this experiment and only concentrations deemed to be non-cytotoxic for each inhibitor were used (Supplementary Fig. 2c). A549 cells were pretreated with Pyr-41 (75 μM), DBeQ (1 μM), EerI (1 μM), ML364 (2 μM) and MG132 (20 μM) for 2 h prior to ZIKV PE243

**Fig. 3 ZIKV C interacts with TER94 and Rpn1 during ZIKV infection in mosquito cells. a** Schematic of cycloheximide chase assay of AF5-V5-C cell line under knockdown conditions with cell lysates obtained at 0, 4, and 8 h post-cycloheximide treatment. **b** Densitometry of $n = 3$ independent cycloheximide chase assay immunoblots of AF5-V5-C cells under TER94 and Rpn1 knockdown conditions. Improved V5-C stability versus dseGFP controls was observed with band intensities (γ-tubulin loading control) presented as mean ± SEM relative signal to 0 h timepoint. **p-value < 0.01 determined by two-tailed two-way ANOVA with Tukey's multiple comparisons test where dsTER94 vs. dseGFP at 4 h $p < 0.001$, 8 h $p < 0.001$; dsRpn1 vs. dseGFP at 4 h $p < 0.001$, 8 h $p < 0.001$; dseGFP 0 vs. 4 h $p < 0.001$, vs. 8 h $p < 0.001$. Exact p-values cannot be computed. Representative blots with anti-V5 and anti-γ tubulin antibodies is presented. **c** Representative immunoblots from $n = 3$ independent co-IP assays showing ZIKV C-TER94 interaction using AF5-V5-C cell line transfected with pPUb-myc-TER94. Immunoprecipitation conducted using anti-V5 or anti-myc antibodies. **d** TER94 and Rpn1 knockdown in AF5 cells and infection for 72 h with ZIKV PE243 (MOI = 1) showed reduced virus replication. Viral RNA was measured 72 hpi by RT-qPCR shown as mean ± SD fold change ($2^{-\Delta\Delta CT}$) relative to S7 gene and dseGFP controls set to 1. Released virus particles were measured by plaque assay with virus titers in PFU/mL (Log10 mean ± SEM). *p-value < 0.05, **p-value < 0.01 determined by two-tailed Student's t-test of $n = 3$ independent repeats for ZIKV PE243 RT-qPCR dsTER94 $p = 0.025$, dsRpn1 $p = 0.005$; for ZIKV PE243 plaque assay dsTER94 $p = 0.040$, dsRpn1 $p = 0.028$. **e** Top panel, schematic of ZIKV NLuc-expressing replicon construct, NLuc reporter followed by last 30 codons of envelope (E30). Bottom panel, replicon assay design in AF5 cells at 24 hpt of dsTER94 and dsRpn1 with ZIKV Replicon RNA co-transfected with pIZ-Fluc plasmid (transfection control). **f** ZIKV Replicon activity following TER94 and Rpn1 knockdown in AF5 cells was measured by dual-luciferase assay (NLuc/FFLuc ratio) expressed as mean ± SEM light units relative to dseGFP controls set to 1 from $n = 3$ independent repeats. *ns = not significant based on two-tailed Student's t-test where dsTER94 $p = 0.638$, dsRpn1 $p = 0.930$. **g** Schematic of ZIKV genome detection assay to measure viral RNA levels internalized into the cell during early stages of infection. At 24 hpt of dsRNA, cells were infected with ZIKV PE243 (MOI = 10) on ice for 30 min to allow simultaneous entry of virus particles. The inoculum was replaced with fresh media with cycloheximide and incubated at 28 °C before total RNA was isolated at 1 and 2 hpi. **h** ZIKV RNA levels shown as mean fold change ($2^{-\Delta\Delta CT}$ ± SD) from CT values normalized to S7 gene and dseGFP controls set to 1 from $n = 3$ independent repeats. *p-value < 0.05 determined by two-tailed Student's t-test compared where dsTER94 vs. dseGFP at 1 hpi $p = 0.016$, 2 hpi 0.276; dsRpn1 vs. dseGFP at 1 hpi $p = 0.484$, 2 hpi $p = 0.952$; and between timepoints (1 vs. 2 hpi) for dsTER94 $p = 0.045$, dsRpn1 $p = 0.756$. Source Data file provided.

infection and the virus was allowed to replicate for 24 h in the presence of inhibitors. It was observed that inhibition of ubiquitination, VCP function, and proteasome activity significantly reduced ZIKV RNA levels versus DMSO controls (Fig. 4d). Inhibition of deubiquitination by ML364 did not affect ZIKV replication.

To assess whether VCP-UPP proteins are needed at certain timepoints during ZIKV infection in A549 cells, a drug time-of-addition assay was designed (Fig. 4e) with ZIKV-NLuc infection. Three conditions were tested: (1) pre-infection wherein cells were treated with inhibitors 2 h prior to infection until 2 hpi; (2) co-treatment wherein ZIKV-NLuc was incubated with inhibitors at 37 °C for 1 h, and the drug-inoculum added to cells for 2 h, and (3) post-infection wherein cells were infected for 2 h before the addition of inhibitors until 6 hpi. Following each treatment, the culture media was replaced with fresh media and incubated for 24 h. Overall, the experiment showed that NLuc signal was reduced only when cells were pre-treated with Pyr-41, DBeQ, and EerI 2 h prior to infection versus DMSO-treated controls (Fig. 4f). Treatment with most drugs post-infection also did not affect NLuc levels. Treating cells with MG132 pre- or post-infection, but not in the case of co-treatment, reduced ZIKV levels, which implies that MG132 also affected virus replication steps post-entry. Comparable to what has been observed before (Fig. 4d), treatment with ML364 did not affect ZIKV replication. We assume that VCP-UPP acts on ZIKV early during the infection of A549 cells, similar to what has been observed with TER94 in AF5 cells. To validate this further, corresponding drug treatments at 2 hpi with ZIKV PE243 were performed, and viral RNA levels measured. Similar results were observed (Supplementary Fig. 2d).

We also performed a time course infection assay to detect the hampered start of infection in earlier timepoints. Indeed, results showed that inhibiting VCP using DBeQ or EerI resulted in a lower NLuc signal compared to DMSO control (with cycloheximide as positive control) at 4 hpi and this difference persisted until 24 hpi (Supplementary Fig. 2e). These data further support the finding that ZIKV capsid interacts with TER94/VCP, and its segregase function is critical in establishing ZIKV replication early in the virus life cycle. In summary, data from both mosquito and human cell experiments showed that the knockdown of TER94/VCP and drug-mediated inhibition of VCP during ZIKV

infection resulted in reduced replication. This suggests a conserved role for TER94/VCP during ZIKV infection in both mosquito and human cells.

**UBR5, a TER94/VCP co-factor needed during ZIKV infection.** It has been discussed that TER94/VCP function relies on co-factors to provide specificity during binding and direct the fate of target proteins; hence it was essential to identify potential co-factors that interact with TER94/VCP during ZIKV infection. To do this, data mining of VCP co-factors was performed in Bio-GRID. As of writing, the database catalogued 953 human proteins that potentially interact with VCP. These were cross-referenced to 151 human orthologs of our *Ae. aegypti* interactome data, resulting in 33 proteins in common (Fig. 5a). GO analysis showed that most proteins function as ribosomal components for protein and RNA binding. Therefore, the interactors were mapped across orthologs to identify potential TER94/VCP co-factors involved during ubiquitination (Fig. 5b). A PPI network of the cross-referenced orthologs involved in ubiquitination was constructed showing 5 human proteins that were highly conserved across species (Fig. 5c). Interestingly, only two proteins function as ubiquitin-binding partners (GO:0043130) across insect species, sequestosome 1 (SQSTM1/p62) and UBR5. SQSTM1 and UBR5 have *D. melanogaster* orthologs, Ref(2)P and Hyd, respectively. *Ae. aegypti* counterparts of these proteins in our proteomics data are AAEL026751 and AAEL020992, respectively.

Little is known about Ref(2)P in mosquitoes regarding its physiological or antiviral function but in *D. melanogaster* it has been shown to limit ZIKV infection[95]. Its human ortholog, SQSTM1, has been linked to cellular regulatory pathways by acting as an adaptor to VCP during autophagy and proteasomal degradation[45,96]. SQSTM1 has been shown to affect flavivirus replication where deficient levels limited RNA replication of Japanese encephalitis virus (JEV), while overexpression suppressed DENV replication[97,98] in mammalian cells. *Ae. aegypti* SQSTM1 (AAEL026751) knockdown did not affect ZIKV replication as shown in our dsRNA knockdown screen (Fig. 2b). To determine if this finding was specific for mosquito cells, siRNAs against SQSTM1 (siSQSTM1-A, siSQSTM1-B, or combined) were tested in A549 cells (Supplementary Fig. 2a, b). Similarly, to what was observed in AF5 cells, SQSTM1 silencing

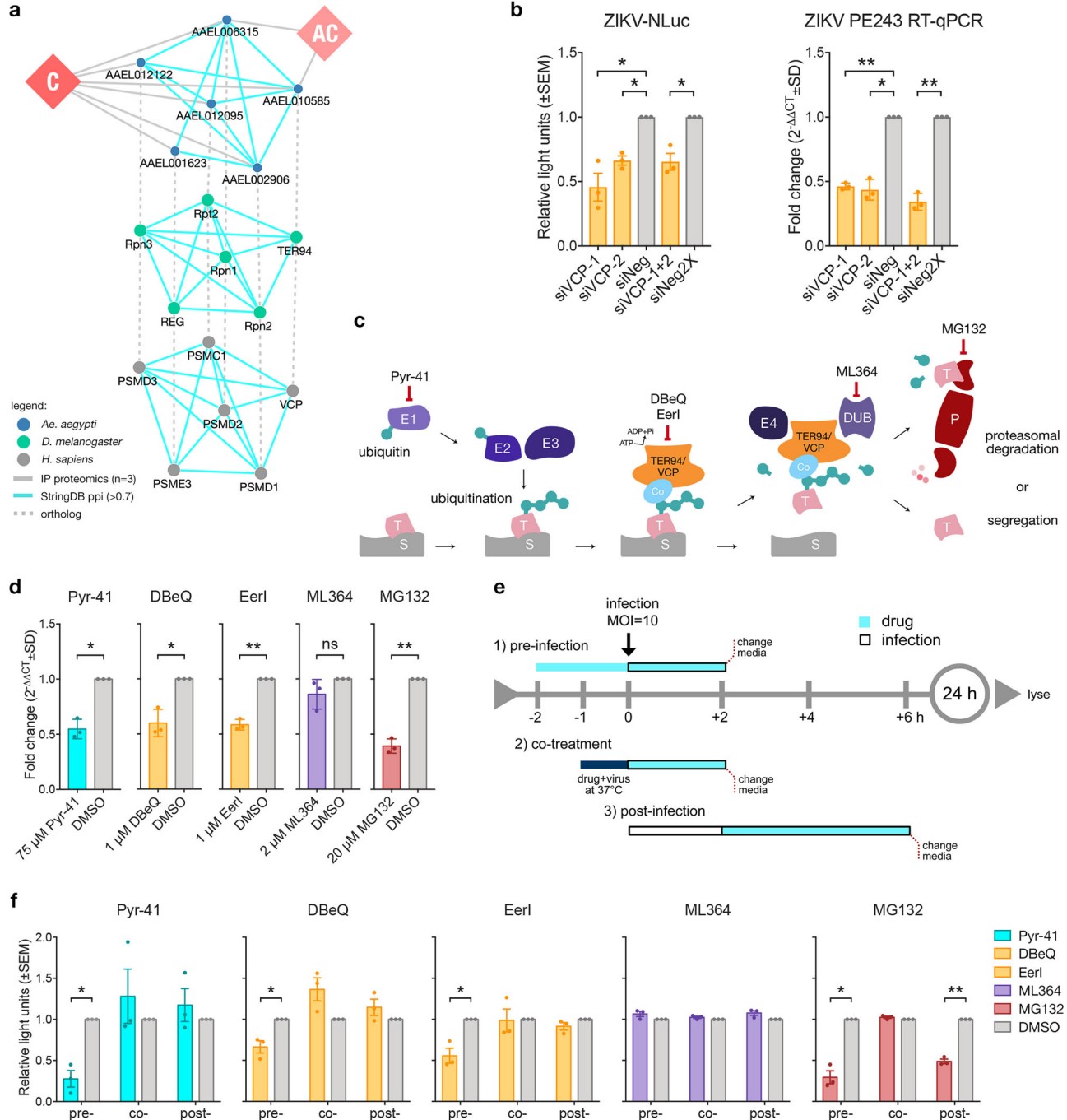

in A549 cells did not affect ZIKV replication (Fig. 5d, left panel), unlike the previous reports involving JEV and DENV.

The other potential TER94/VCP co-factor, UBR5 is an E3 ubiquitin ligase that recognizes protein degradation signals or degrons enabling K11/K48 ubiquitination of internal lysine residues[99–101]. The role of UBR5 in oncogenesis has been reviewed extensively highlighting its function in cellular development and cell cycle regulation[102]. A recent study describes the interaction of UBR5 with VCP, leading to Wnt signaling pathway inactivation[103]. Initially, AeUBR5 was not included in the dsRNA knockdown screen as although it was identified in three AC samples, it only appeared in two C IP proteomics samples. To test the role of UBR5 during ZIKV infection, siRNAs against UBR5 (siUBR5-A, siUBR5-B, or combined) were transfected into A549 cells (Supplementary Fig. 2a, b). Following infection with ZIKV-

NLuc, silencing of UBR5 resulted in significantly reduced NLuc readings relative to siNeg controls (Fig. 5d, right panel).

Given the results in human cells, it was of interest to know if *Ae. aegypti* UBR5 (AAEL020992; AeUBR5) affected ZIKV replication. Infection experiments in AF5 cells transfected with dsAeUBR5 were performed. Knockdown efficiency of dsAeUBR5 was checked by RT-qPCR and was not found to affect cell viability (Supplementary Fig. 1f). Infecting with either ZIKV-NLuc (MOI = 1) or ZIKV PE243 (MOI = 1) for 72 h showed significantly reduced virus replication versus controls, with less infectious virus released as shown by ZIKV PE243 plaque assay done in parallel (Fig. 5e).

To provide evidence to the interaction of TER94 and AeUBR5 in relation to ZIKV C, co-IP assays in AF5-V5-C cells under AeUBR5 knockdown conditions were carried out. Briefly, stable

**Fig. 4 Ortholog mapping of TER94 and proteasome subunits reveal conserved host factors during ZIKV infection in human cells. a** ZIKV capsid-*Ae. aegypti* TER94 and proteasome subunits PPI (blue nodes) from our proteomics data (gray edges) with *D. melanogaster* (green nodes) and *H. sapiens* (gray nodes) orthologs. Cyan edges are StringDB PPI. Orthologs connected by dashed lines. **b** VCP silencing in A549 cells with siRNAs (siVCP-1 or siVCP-2 compared to siNeg; siVCP-1 and siVCP-2 (siVCP-1 + 2) compared to double amounts of negative siRNAs (siNeg2X) and infected with ZIKV-NLuc (MOI = 10) or ZIKV PE243 (MOI = 10) 72 hpt for 24 h reduced ZIKV levels. Results from $n = 3$ independent repeats shown as mean ± SEM light units or mean ± SD fold change ($2^{-\Delta\Delta CT}$) of CT values normalized to GAPDH and siNeg controls set to 1. *$p$-value < 0.05, **$p$-value < 0.01 determined by two-tailed Student's $t$-test for ZIKV-NLuc siVCP-1 $p = 0.037$, siVCP-2 $p = 0.011$, siVCP-1 + 2 $p = 0.033$; for ZIKV PE243 RT-qPCR siVCP-1 $p = 0.002$, siVCP-2 $p = 0.015$, siVCP-1 + 2 $p = 0.009$. **c** A general model of VCP-UPP in human cells showing different steps from ubiquitination of target proteins (T) on a substrate (S). Segregase function of VCP with a co-factor (Co) leads to proteasomal (P) degradation. Chemical inhibitors of UPP used in this study and their stage of action are indicated. **d** Treating A549 cells with chemical inhibitors 2 h pre-infection of ZIKV PE243 (MOI = 10) reduced viral RNA levels versus DMSO controls when ubiquitination (Pyr-41), VCP (DBeQ and EerI), and proteasome (MG132) activities were suppressed. $N = 3$ independent repeats shown as mean fold change ($2^{-\Delta\Delta CT}$ ± SD) of CT values normalized to GAPDH and DMSO controls set to 1. ns = not significant, *$p$-value < 0.05, **$p$-value < 0.01 determined by two-tailed Student's $t$-test where Pyr-41 $p = 0.021$; DBeQ $p = 0.044$; EerI $p = 0.007$; ML364 $p = 0.237$; MG132 $p = 0.009$. **e** Time-of-addition assay of chemical inhibitors during ZIKV-NLuc (MOI = 10) infection in A549 cells. Drug treatments done: 1) pre-infection [pre-]; 2) co-treatment [co-], and 3) post-infection [post-]. **f** Treatment of A549 with Pyr-41, DBeQ, EerI, MG132 at 2 h pre-infection reduced virus levels versus DMSO controls. $N = 3$ independent repeats presented as mean ± SEM light units relative to DMSO control set to 1. *$p$-value < 0.05, **$p$-value < 0.01 determined by two-tailed Student's $t$-test where pre- Pyr-41 $p = 0.019$, DBeQ $p = 0.046$, EerI $p = 0.039$, MG132 $p = 0.012$; for post- MG132 $p = 0.003$; other $p$-values provided in the Source Data file.

cells were transfected with myc-TER94 expressing plasmids for 48 h and transfected with dsAeUBR5 or dsLacZ controls. Assays were performed with MG132 and ML364 as previously mentioned to limit the effect of degradation. Immunoblots showed that knockdown of AeUBR5 resulted in the loss of interaction between ZIKV C and TER94 (Fig. 5f).

The effect of AeUBR5 silencing on ZIKV C protein degradation was also investigated. A cycloheximide chase assay was performed as above under AeUBR5 knockdown conditions in the AF5-V5-C cell line. Silencing AeUBR5 improved V5-C protein stability compared to controls, with minimal degradation within the 8 h of cycloheximide treatment (Fig. 5g). This result is comparable to the effect of silencing TER94 on V5-C stability (Fig. 3d). We conclude that less ZIKV C was targeted for degradation when AeUBR5 was silenced. Furthermore, AeUBR5 knockdown also does not affect ZIKV Replicon signal, similar to TER94 silencing (Fig. 5h). Overall, this suggests a pro-viral role for UBR5 during ZIKV infection concurrent with previous results observed with TER94. These data corroborate the hypothesis that ZIKV capsid can be degraded by the proteasome through TER94/VCP segregation with UBR5 as a co-factor.

## Discussion

Here, we present a method for studying *Ae. aegypti*-virus PPIs using stable mosquito cell lines expressing virus proteins. These were generated in a mosquito cell line expression system we had previously developed[66]. Using the AF5 cell line, a single cell-derived clone from Aag2 cells, allows future work on the identified interactors to be done using CRISPR-Cas9 knockout or knock-in experiments[66]. In addition, AF5 cells exhibit better transgene expression without sacrificing growth kinetics and arbovirus infectivity[104], making it well suited for molecular experiments and developing stable mosquito cell lines.

This system can be easily adapted to investigate further ZIKV proteins, or those of other mosquito-borne arboviruses, as well as endogenous mosquito proteins. With a pipeline of investigation that utilizes proteomics data, network analyses, bioinformatics tools, and in vitro experiments, we were able to identify and test pro-viral mosquito proteins that have orthologs in humans. We provide new data and implications relating to how ZIKV infection occurs in vector cells and extrapolate the key findings to human cells. Although, a number of proteomics-based studies on arbovirus-vector interactions have been conducted in cells and whole organisms[105–108], our cell line system opens up the potential to study more pronounced PPIs as it captures a global interactome not restricted by molecular changes caused by virus replication cycles where transient interactions are often lost. We acknowledge that some interactions may only take place in the context of virus infections and would not be captured here. However, this system provides a toolkit approach for the study of individual, or combinations of viral proteins at the cellular level. This makes it highly flexible, robust, and complementary to recent approaches to PPI studies of ZIKV infection in vertebrate cells[36,37]. It may also be used as an alternative to screening approaches using Y2H libraries[42] or *D. melanogaster* systems that have been used, for example, to identify TER94/VCP interaction with SINV[87]. Moreover, it provides a more vector-focused approach to investigating PPI as shown by the very minimal overlap between previous PPI and genome-wide screen studies.

By using gene silencing and pharmacological approaches to dissect ZIKV-TER94/VCP interactions, we were able to show how vital these interactions are in establishing infection in both mosquito and human cells. This interaction also expands our understanding of mosquito UPP where a novel ubiquitin that targets DENV E for degradation was recently discovered[109]. Notably, VCP has never been implicated in other interactome studies involving ZIKV proteins in human cells[35–37], but has been recently identified to interact with ZIKV NS4B in forming virus replication factories[110]. We propose a model where TER94/VCP, along with UBR5 as a co-factor, is critical for ZIKV infection post-fusion, where capsid bound to genomic RNA is ubiquitinated following the UPP model (Fig. 6). UBR5 directs TER94/VCP interaction with C, eventually orchestrating the disassembly of nucleocapsid structures, thereby exposing the viral RNA genome to the cytoplasm. As a consequence, C can be degraded by proteasomal complexes. This hypothesis corroborates structural studies on ZIKV capsid and genome encapsulation[27,28]. Although we do not yet fully understand the detailed mechanism of TER94/VCP interaction with capsid and the role of UPP proteins, including UBR5, our data provide further proof that ubiquitination of viral proteins is essential during infection. Diving deeper into these interactions with biochemical approaches will be necessary to investigate the underlying mechanisms involved.

VCP has been hypothesized to play a role in the ubiquitination of DENV C and subsequent uncoating of DENV genome in human cells[75]. It was also recently investigated during YFV infection[91]. Although a different drug-screening-based approach was used in mammalian cells, the results of our study support their findings, critically linking *Ae. aegypti* TER94 to flavivirus capsid and other UPP components. Although VCP interaction

 

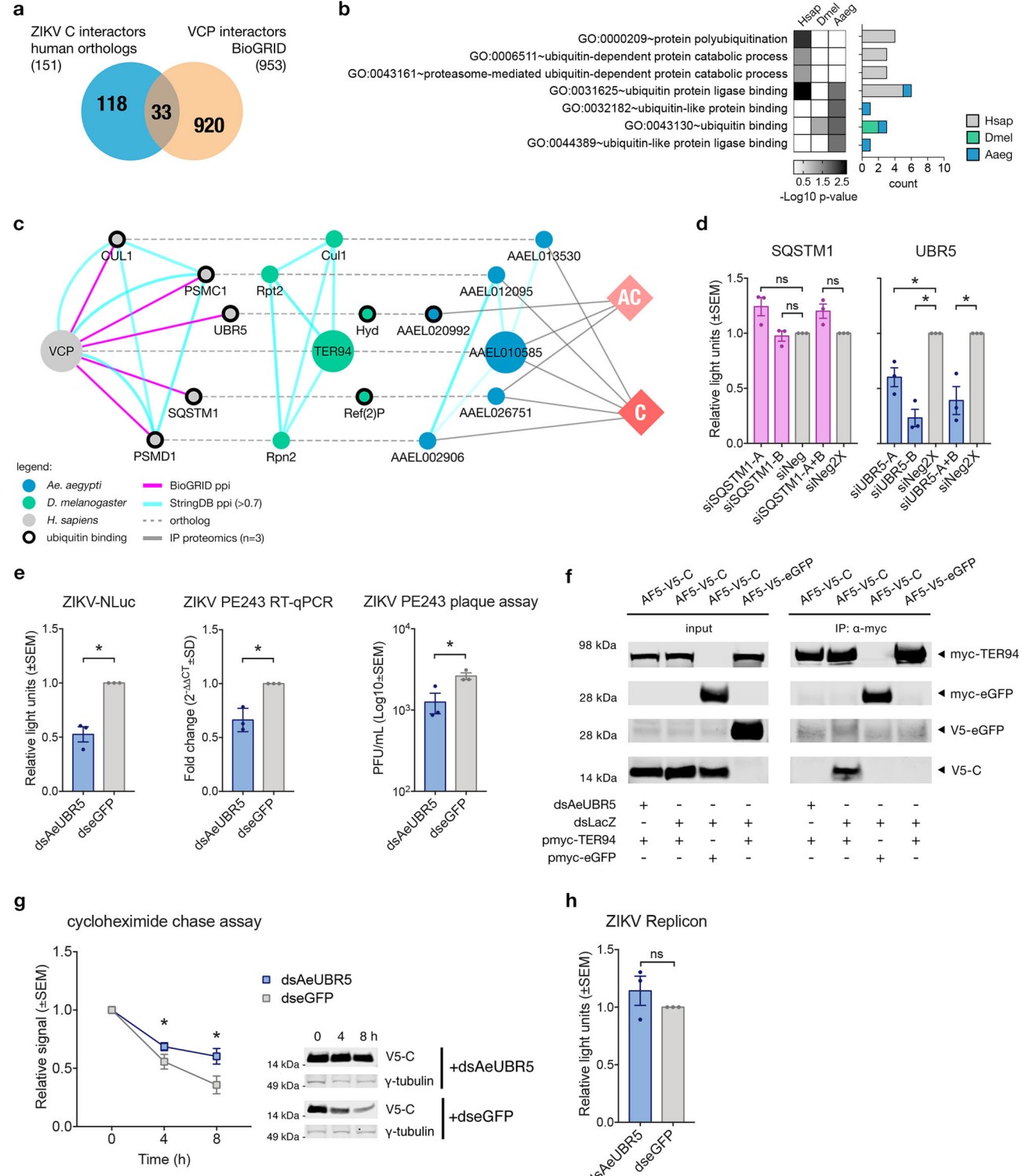

with viruses has been explored widely, most studies involved mammalian systems[85,86,88,89,111]. Here, we show how its *Ae. aegypti* ortholog, TER94, behaves similarly during ZIKV infection in vector cells, improving our understanding of common and diverging aspects of infection across host cells.

The UPP is a multistep process with proteasomal degradation as the end result, and we have shown that the function of UBR5 and TER94 may be a rate-limiting step during the early stages of ZIKV infection. Although a proteomics study involving ZIKV infection of C6/36 cells suggested that proteasome activity was

deemed important for virus entry[108], it has also been shown to be important in DENV egress[112] and virion production[77]. In addition, the UPP has been hypothesized to be important at early stages of JEV[94], porcine circovirus[113], murine coronavirus[114], and influenza A virus[115] infections. These reports show the diversity of roles the UPP plays during virus infections and is further supported by what we have observed in our results.

This is to our knowledge, the first time UBR5 has been linked to TER94/VCP as a co-factor in its interaction with a virus. Previous studies have described UBR5 function in regulatory

**Fig. 5 UBR5 is a TER94/VCP co-factor during ZIKV infection. a** Overlap of BioGRID VCP interactors and human orthologs of *Ae. aegypti* ZIKV capsid interactors. **b** GO analysis showing cross-referenced VCP interactors from BioGRID involved in ubiquitination across *H. sapiens* (Hsap), *D. melanogaster* (Dmel), and *Ae. aegypti* (Aaeg). The heat map indicates significance values (in −Log10 *p*-value) of only significant enrichments (*p*-value < 0.05) with Benjamini correction against the background gene set with actual *p*-values provided in the Source Data file. Bars indicate gene count. **c** Ortholog-PPI network of VCP interactors involved in ubiquitination from BioGRID. *H. sapiens*, *D. melanogaster*, and *Ae. aegypti* proteins are gray, green, and blue nodes, respectively. BioGRID interaction edges in magenta. StringDB PPI in cyan edges. Orthologs are connected by dashed lines. Mosquito interactors to ZIKV C and AC are linked by gray edges. **d** siRNA silencing of UBR5 and SQSTM1 in A549 cells for 72 h prior to infection with ZIKV-NLuc (MOI = 10) for 24 h. UBR5 knockdown resulted in reduced NLuc levels versus controls with NLuc readings shown as mean ± SEM light units relative to siNeg controls set to 1 from *n* = 3 independent experiments. ns = not significant, *$p$-value < 0.05 determined by two-tailed Student's *t*-test where siSQSTM1-A $p = 0.099$; siSQSTM1-B $p = 0.642$; siSQSTM1-A + B $p = 0.091$; siUBR5-A $p = 0.043$; siUBR5-B $p = 0.010$; siUBR5-A + B $p = 0.041$. **e** In AF5 cells, AeUBR5 knockdown using dsRNA and infection with ZIKV-NLuc (MOI = 1) or ZIKV PE243 (MOI = 1) at 24 hpt. ZIKV levels at 72 hpi showed reduced NLuc signal and viral RNA versus dseGFP controls. ZIKV PE243 plaque assay also showed reduced titers in dsAeUBR5. *N* = 3 independent experiments presented as mean ± SEM light units or fold change ($2^{-\Delta\Delta CT} \pm$ SD) with CT values normalized to S7 gene relative to dseGFP controls set to 1. Virus titers in PFU/mL (Log10 mean ± SEM). *$p$-value < 0.05 determined by two-tailed Student's *t*-test for ZIKV-NLuc $p = 0.021$; ZIKV PE243 RT-qPCR $p = 0.040$; ZIKV PE243 plaque assay $p = 0.045$. **f** Representative immunoblots from *n* = 3 independent co-IP assays performed in AF5-V5-C cells under AeUBR5 knockdown conditions and transiently expressing myc-TER94 was performed. Knockdown of AeUBR5 resulted in the loss of V5-C pulldown by anti-myc antibody. **g** Densitometry of *n* = 3 independent cycloheximide chase assay immunoblots in AF5-V5-C cells at 24 hpt of dsAeUBR5. Band intensities (γ-tubulin loading control) shown as mean ± SEM signal relative to 0 h timepoint. *$p$-value < 0.05 determined by two-tailed two-way ANOVA with Tukey's multiple comparisons test where dsAeUBR5 vs. dseGFP at 4 h $p = 0.263$, 8 h $p = 0.015$; dsAeUBR5 at 0 vs. 4 h $p = 0.002$, vs. 8 h $p < 0.001$; for dseGFP at 0 vs. 4 h $p < 0.001$, vs. 8 h $p < 0.001$. Exact $p$-values cannot be computed. **h** AF5 cells under AeUBR5 knockdown conditions were transfected with in vitro transcribed ZIKV Replicon RNA together with pIZ-FLuc, and at 24 hpt the signal intensities were measured by dual-luciferase assay. NLuc/FFLuc ratio expressed as mean ± SEM light units relative to dseGFP controls set to 1 from *n* = 3 independent experiments. ns = not significant determined by two-tailed Student's *t*-test where $p = 0.379$. Source Data file is provided.

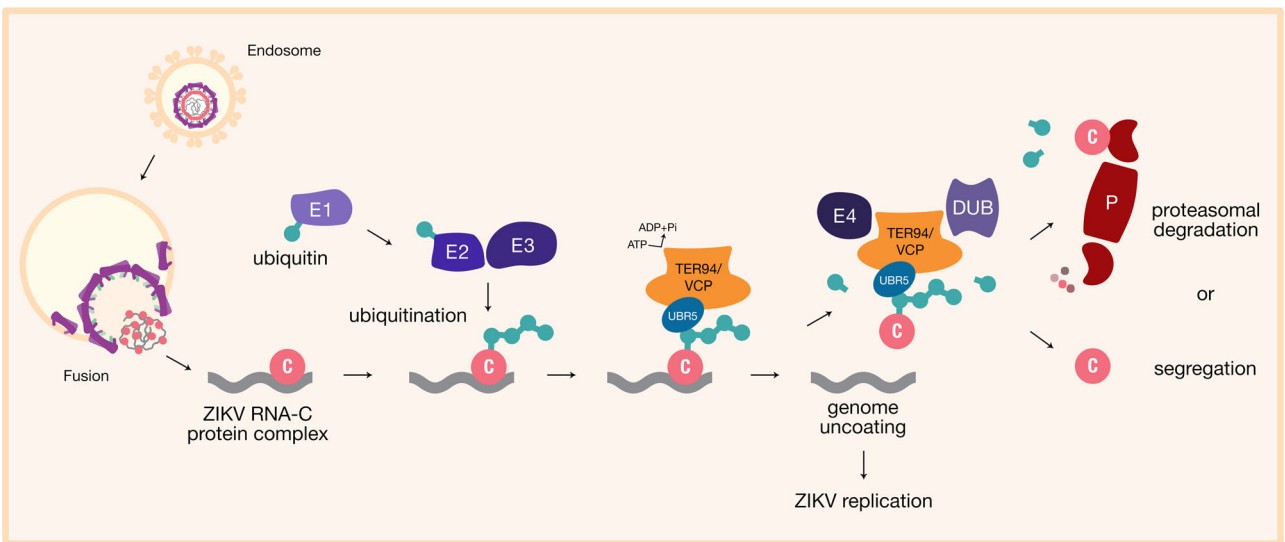

**Fig. 6 Proposed model of TER94/VCP-mediated ZIKV uncoating.** Ubiquitination of ZIKV C by ligases (E1, E2, and E3) allows interaction with UBR5 leading to the involvement of VCP/TER94. Segregation allows disassembly of the nucleocapsid, uncoating the RNA genome into the cytoplasm for translation. Degradation of ZIKV C by proteasomal complexes (P) can be a by-product of this interaction.

pathways and cancer[100,102,103] but not much is known about how it functions in tandem with TER94/VCP during viral infections. This merits further investigation. Altogether, this study adds novel information on the early stages of ZIKV-vector/host cell interactions, with key aspects conserved in arthropod and vertebrate cells, and it will be of importance to the field to compare this to other flaviviruses.

## Methods

**Cells.** *Ae. aegypti* AF5 cells, clonally derived from Aag2 cells[66,104] were previously used for CRISPR-Cas9 knockouts experiments. The AF5 cells were grown in L-15 medium with GlutaMAX (Thermo Fisher Scientific) supplemented with 10% tryptose phosphate broth (TPB; Thermo Fisher Scientific), 10% fetal bovine serum (FBS; Thermo Fisher Scientific), and penicillin-streptomycin (100 units/mL and 100 μg/mL, respectively, Thermo Fisher Scientific) at 28 °C with no added $CO_2$. Stable cell lines (AF5-V5-C, AF5-V5-AC, and AF5-V5-eGFP) were also grown in the same conditions as stated with the addition of Zeocin (1 g/mL; Invivogen).

A549 cells (human male Caucasian lung carcinoma cells; ECACC, Cat# 86012804) were grown in DMEM (Thermo Fisher Scientific) supplemented with 10% FBS at 37 °C with 5% $CO_2$. A549 cells stably expressing bovine viral diarrhea virus NPro[116] cells (A549-Npro) were a kind gift from R.E. Randall (University of St Andrews, UK) were grown in similar conditions as A549 cells but with the addition of Blasticidin (100 mg/mL, Invivogen) to keep the transgene. Cell lines made for the study are available upon request.

**Viruses.** ZIKV-NLuc reporter virus was rescued from a ZIKV icDNA clone, pCCI-SP6-ZIKV-NLuc based on a Brazilian ZIKV isolate[48]. ZIKV PE243 (ZIKV/H. sapiens/Brazil/PE243/2015; GenBank: KX197192.1; https://www.ncbi.nlm.nih.gov/nuccore/KX197192.1) is a Brazilian clinical isolate from a patient, which we have previously characterized[76]. All viruses were grown in A549-NPro cells in DMEM with 2% FBS at 37 °C with 5% $CO_2$ for five days. Culture supernatants were collected and centrifuged at $443 \times g$ for 5 min. Virus titers were determined by plaque assay and aliquots stored at −80 °C.

**Plasmids.** Plasmids used in mosquito cells where the expressed genes are placed were under the control of a polyubiquitin promoter (PUb) from pGL3-PUb[49]

(Addgene, Cat# 52891) originally expressing FFLuc. The luciferase coding sequence was removed, leaving a PUb backbone plasmid (called pPUb), which was used to generate a plasmid expressing V5-eGFP (pPUb-V5-eGFP[66]). pPUb-V5-eGFP contains a ZeoR gene, two copies of 2 A autoprotease sequence of Thosea asigna virus, and a V5-tag. This was used as the backbone to generate constructs expressing V5-tagged ZIKV C/AC by replacing the eGFP coding sequence. The eGFP coding sequence was removed by restriction enzyme digestion using XhoI (New England BioLabs, Cat# R0575S) and BsrGI (New England BioLabs, Cat# R0146S) with CIP (New England Biolabs, Cat# M0525S). Coding sequences for C (315 bp) or AC (369 bp) were PCR amplified from pCCI-SP6-ZIKV using gene-specific primers (ZCFor with ZIKVAC_R or ZIKVC_R primers, Supplementary Table 4) with 15 bp homologous overlaps to the backbone using CloneAmp HiFi PCR Premix (Takara Bio). The sequence is identical to C or AC in ZIKV isolate PE243[76]. For the myc-tagged TER94 expression plasmid (pPUb-myc-TER94), a pPUb-myc-eGFP plasmid previously made[66] was used as the backbone. eGFP coding sequence was removed using restriction with XhoI and FseI (New England BioLabs, Cat# R0588S) in the presence of CIP. TER94 coding sequence was PCR amplified from reverse-transcribed total RNA extracted from AF5 cells using gene-specific primers (XhoI_TER94_F with TER94_FseI_R primers, Supplementary Table 4) with homologous overlaps and CloneAmp HiFi PCR Premix. Each amplicon of interest was gel purified and subjected to In-Fusion cloning (Takara Bio) following the manufacturer's protocol. The mix was then transformed in Stellar Competent Cells (Takara Bio, Cat# 636763) and grown on LB agar with ampicillin (100 μg/μL) for 24 h at 37 °C. Sequences of inserts in bacterial colonies were confirmed using Sanger sequencing. DNA preparations used in the studies were made using PureLink HiPure Plasmid Maxiprep kit (Thermo Fisher Scientific) as per manufacturer's guidelines. Sequences and primers analyzed in Geneious Prime 2021.0.3.

**Antibodies**. Primary antibodies: mouse monoclonal anti-V5 tag antibody (Abcam, Cat# ab27671) was used for IP, co-IP, immunoblotting, and immunofluorescence assays; mouse monoclonal anti-myc tag antibody (Abcam, Cat# ab32) was used for co-IP assays and immunoblotting; mouse monoclonal anti-UBR5 (Proteintech Group, Cat# 66937-1-Ig) and anti-β catenin (Cell Signaling Technology, Cat# 2677 s) were used for immunoblotting; and rabbit polyclonal anti-VCP (Proteintech Group, Cat# 10736-1-AP), anti-SQSTM1 (Proteintech Group, Cat# 18420-1-AP), anti-V5 tag (Abcam, Cat# ab9116), anti-γ-tubulin (Abcam, Cat# ab11317), and anti-β catenin (Abcam, Cat# ab16051) antibodies were used for immunoblotting. Secondary antibodies for immunoblotting: goat polyclonal anti-mouse HRP-conjugated (Thermo Fisher Scientific, Cat# A16072), anti-mouse IgG (H + L) DyLight 800 (Thermo Fisher Scientific, Cat# SA5-35521), and anti-rabbit IgG (H + L) DyLight 680 (Thermo Fisher Scientific, Cat# 35568). Secondary antibodies for immunofluorescence microscopy: goat anti-mouse Alexa Fluor 568 (Thermo Fisher Scientific, Cat# A-11031).

**Development of stable mosquito cell lines**. To generate the stable cell lines, 20 μg of pPUb-V5-eGFP, pPUb-V5-C or pPUb-V5-AC were initially linearized using NotI-HF (New England BioLabs, Cat# R3189S) restriction enzyme at 37 °C for 24 h and purified using QIAquick PCR purification kit. AF5 cells seeded at $2.5 \times 10^5$ cells/well in T25 flasks were transfected with 5 μg of the linearized plasmid using 10 μL DharmaFECT 2 (Horizon Discovery) in Opti-MEM (Thermo Fisher Scientific). After 24 h, the culture media was replaced with fresh L-15 supplemented with Zeocin (1 g/mL at 1:1000, Invivogen) and thereafter replaced every 4 days for 30 days. An aliquot of cells was obtained 30 days post-transfection and lysed for immunoblotting of V5 tag. The remaining cells were then split and maintained in L-15 supplemented with Zeocin (1 g/mL at 1:500).

**Cycloheximide chase assay**. To evaluate ZIKV C degradation during knockdown conditions of TER94, Rpn1, and AeUBR5, AF5-V5-C cells seeded at $2.5 \times 10^5$ cells/well in 24-well plates were transfected correspondingly with 300 ng dsRNA or dseGFP control with 2 μL DharmaFECT 2 (Horizon Discovery) in Opti-MEM. Cycloheximide chase assay was performed 24 hpt by adding cycloheximide (1 μg/μL, Sigma-Aldrich) for 0, 4, and 8 h.

**IP and co-IP assays**. For IP samples for proteomics, AF5 stable cell lines grown in T150 flasks were scraped and spun at $443 \times g$ for 5 min to pellet the cells. The supernatant was removed, and the cell pellet washed with phosphate-buffered saline (PBS; Thermo Fisher Scientific) and spun down. Cells were then resuspended in IP lysis buffer (20 mM HEPES [pH 7.4]; Thermo Fisher Scientific], 150 mM NaCl, 5 mM MgCl$_2$, 1× Halt protease inhibitor cocktail [1:100 dilution; Thermo Fisher Scientific], and 1% Triton X-100) and incubated on ice for 30 min. Lysed cells were then centrifuged at $16,000 \times g$ for 20 min at 4 °C. The supernatant was incubated with mouse monoclonal anti-V5 tag antibody (1:200 dilution; Abcam, Cat# ab9116) for 2 h at 4 °C on a rotator disk. Protein G Dynabeads (Thermo Fisher Scientific) were equilibrated by washing with cold IP wash buffer (same as the IP lysis buffer without protease inhibitors) for at least an hour at 4 °C on a rotator disk. To the supernatant-antibody mix, 30 μL pre-equilibrated Protein G Dynabeads was then added and incubated for another hour at 4 °C on a rotator disk. The slurry was then placed on a magnetic rack and washed three times with

IP wash buffer for 5 min and transferred to a fresh tube. The beads were finally resuspended in 50 μL IP wash buffer, 20 μL 4× Bolt LDS (Thermo Fisher Scientific), and 10 μL 10× Bolt Reducing Agent (Thermo Fisher Scientific). An aliquot was used for immunoblotting. Independent triplicate IP assays were done for proteomics analysis. For co-IP, AF5-V5-C cells were seeded into T150 flasks and allowed to adhere overnight. The cells were then transfected with 30 μg of pPUb-myc-TER94 with 15 μL DharmaFECT 2 in Opti-MEM. After 72 h, transfected cells were treated with 10 μM MG132 (Merck) and 10 μM ML364 (Sigma-Aldrich) for 1 h prior to performing the IP assay mentioned earlier. IP lysis and wash buffers were modified by adding 10 μM MG132 and 10 μM ML364. For the co-IP assay under AeUBR5 knockdown conditions, stable cells transiently expressing myc-TER94 were transfected with 300 ng dsAeUBR5 or dsLacZ at 24 h prior to lysis. A reciprocal pulldown assay of myc-TER94 was also performed using mouse anti-myc tag antibody (1:200; Abcam, Cat# ab32) to co-immunoprecipitate V5-C. Input cell lysate and IP samples were then probed for either V5-C or myc-TER94.

**Immunoblotting**. Samples lysed using 1× Bolt LDS Sample Buffer with 1× Bolt Reducing Agent (Thermo Fisher Scientific) were heated at 95 °C for 10 min. Samples were then separated on Bolt 4-12% Bis-Tris Plus gels (Thermo Fisher Scientific) and transferred to 0.45 μm nitrocellulose membrane (Thermo Fisher Scientific) using a Trans-Blot SD Semi-Dry Transfer Cell (Bio-Rad). Membranes were then blocked with 5% (w/v) non-fat dry milk in PBS-Tween (PBS with 0.1% Tween 20) for at least 1 h. Membranes were then incubated with primary antibodies in 2% (w/v) non-fat dry milk in PBS-Tween for 1 h using mouse monoclonal anti-V5 tag antibody (1:2000, Abcam, Cat# ab27671), mouse monoclonal anti-myc tag antibody (1:2000, Abcam, Cat# ab32), mouse monoclonal anti-UBR5 (1:2000, Proteintech Group, Cat# 66937-1-Ig), rabbit polyclonal anti-VCP (1:2000, Proteintech Group, Cat# 10736-1-AP), anti-SQSTM1 (1:2000, Proteintech Group, Cat# 18420-1-AP) or anti-V5 tag (1:2000, Abcam, Cat# ab9116). To detect housekeeping proteins in the same gel, rabbit anti-γ-tubulin (1:2000, Abcam, Cat# ab11317), rabbit anti-β catenin (1:2000, Abcam, Cat# ab16051), or mouse anti-β catenin (1:2000, Cell Signaling Technology, Cat# 2677 s) were used, when appropriate. Membranes were then washed three times with PBS-Tween for 10 min. In case of the IP samples, membranes were incubated with polyclonal goat anti-mouse HRP-conjugated secondary antibody (1:5000; Thermo Fisher Scientific, Cat# A16072) for 1 h and washed three times with PBS-Tween for 10 min. Membranes were then incubated in Amersham ECL Western Blotting Detection Reagent (Merck, Cat# GERPN2109) and chemiluminescent bands detected using Gel Doc XR+ with Image Lab software (v.4.1; Bio-Rad). For cycloheximide chase assay and co-IP samples, membranes were incubated with goat anti-mouse DyLight 800 (1:5000; Thermo Fisher Scientific, Cat# SA5-35521) and/or goat anti-rabbit DyLight 680 (1:5000; Thermo Fisher Scientific, Cat# 35568) secondary antibody conjugated with a near fluorescent dye in 2% (w/v) non-fat dry milk in Tween-PBS for 1 h. Membranes were again washed with Tween-PBS three times and viewed using an Odyssey CLx with Image Studio (v.1.0.11; LI-COR Biosciences). Densitometry analyses of immunoblot images performed in Image Studio Lite (v.5.2.5; LI-COR Biosciences).

**Immunofluorescence analysis**. Immunostaining of AF5-V5-C or AF5-V5-AC cells was performed by seeding $1.5 \times 10^5$ cells/well into poly-D-lysine treated 13 mm coverslips in 24-well plates. After 24 h cells were fixed with 4% formaldehyde for 20 min, washed with PBS for 5 min, and permeabilized using 0.5% Triton X-100 in PBS for 10 min. After another PBS wash, coverslips were blocked with 5% FBS in PBS for 1 h. Detection of V5-C or V5-AC was done for 1 h using monoclonal mouse anti-V5 tag (1:500) antibody in 5% FBS in PBS. Coverslips were then washed three times for 5 min with PBS and incubated with goat anti-mouse Alexa Fluor 568 (1:1000; Thermo Fisher Scientific, Cat# A-11031) in 5% FBS in PBS for 1 h in the dark. Following three washes of PBS for 5 min, coverslips were washed with ddH$_2$O and mounted on glass slides with VECTASHIELD HardSet mounting medium with DAPI (Vector Laboratories, Cat# H-1500). Glass slides were viewed under a Zeiss LSM 710 confocal microscope with ZEN black 2011 version. Photomicrographs were analyzed in ZEN lite (v.3.3; Zeiss).

**LFQ proteomics of IP samples**. All proteomics experiments were performed at the Fingerprints Proteomics Facility (University of Dundee, https://www.lifesci.dundee.ac.uk/technologies/fingerprints-proteomics-facility). Proteins from IP samples were initially separated by SDS-PAGE with gel selection processing, in-gel reduction/alkylation, and in-gel trypsin digestion. Digested samples were then subjected to nLC-MS/MS using an LTQ Orbitrap Velos Pro (Thermo Fisher Scientific). Raw MS data were analyzed in MaxQuant (v1.6.3.4) with a 1% false discovery rate at the protein and peptide spectrum match levels. Treating each raw file as a separate fraction in the experimental design, the following parameters were observed: fixed modifications were carbamidomethyl (C), variable modifications were oxidation (M) and acetyl (protein N-terminal), LFQ and iBAQ were enabled, digestion was with trypsin/P, and default options were used for all other search settings. Peptide spectra were matched to a database containing: (i) the bait protein sequences; (ii) the *Ae. aegypti* reference proteome (UP000008820 downloaded from UniProt[117] using a version last updated 29 June 2020), edited to remove all of the multiple instances of the ubiquitin sequence to prevent ambiguous assignments of

ubiquitin-derived peptide spectra; (iii) a single instance of the ubiquitin sequence; and (iv) the default MaxQuant list of common contaminants. Protein groups matching to these databases were subsequently excluded if they were only identified by site, or if they were found on the common contaminants list (with the exception of eGFP, which was also a bait protein). To select potential interactors, the protein hits were scored based on replicate reproducibility between samples, iBAQ normalization[50], and SAINTq[51]. For further technical details on sample processing and analysis see Data Availability.

**cDNA synthesis, dsRNA synthesis, and RT-qPCR.** Total RNA was extracted from cells (pooled from 4 wells) using TRIzol (Thermo Fisher Scientific) following the manufacturer's instructions with the addition of glycogen during RNA precipitation to improve the yield. A 1 μg aliquot of RNA was used for cDNA synthesis using Superscript III Reverse Transcriptase (Thermo Fisher Scientific) and random primers (Promega, Cat# C1181). Gene-specific primers flanked with T7 RNA polymerase promoters (Supplementary Table 5) and GoTaq G2 Flexi polymerase (Promega) were used to obtain templates for dsRNA synthesis. Amplicons were gel purified using QIAquick Gel extraction kit (Qiagen) with 1 μg used for in vitro transcription with MEGAscript RNAi kit (Thermo Fisher Scientific) following the manufacturer's protocol. For quantitative PCR of ZIKV and genes of interest for knockdown efficiency were performed using gene-specific qPCR primers (Supplementary Table 6) and Fast SYBR Green Master Mix (Thermo Fisher Scientific) in an ABI7500 Fast Real-Time PCR system with 7500 software (v.2.3; Thermo Fisher Scientific) with S7 (for *Ae. aegypti* cells) and GAPDH (for A549 cells) as housekeeping genes.

**dsRNA knockdown and infection in *Ae. aegypti* cells.** For the dsRNA knockdown screen, candidate proteins were chosen based on function, availability of supporting information, known interactions with other viruses, and excluding ribosomal subunits and cytoskeleton proteins (Supplementary Table 3; *Ae. agypti* mRNA sequences from https://www.ncbi.nlm.nih.gov/assembly/ GCF_002204515.2). To perform the dsRNA knockdown screen and TER94, Rpn1, and AeUBR5 silencing, AF5 cells seeded at $2.5 \times 10^5$ cells/well in 24-well plates were transfected with 300 ng of dsRNA or dseGFP as control with 2 μL DharmaFECT 2 (Horizon Discovery) in Opti-MEM. Cells were then infected 24 hpt either with ZIKV-NLuc (MOI = 1) or ZIKV PE243 (MOI = 1) for 72 h. Cells infected with ZIKV-NLuc were lysed with 1× PLB and NLuc readings measured using Nano-Glo Luciferase Assay system (Promega, Cat# N1130) on GloMax Luminometer with Instinct software (v.2.0.1; Promega). For cells infected with ZIKV PE243, total RNA was extracted using TRIzol following the manufacturer's protocol and subjected to RT-qPCR. Parallel knockdown efficiencies of the dsRNAs were also measured by RT-qPCR of gene transcripts and corresponding cell viabilities were checked prior to the actual experimentation using CellTiter-Glo assay (Promega, Cat# G7571).

**ZIKV replicon assay.** A ZIKV derived, NLuc expressing replicon, abbreviated ZIKV Replicon, was designed based on a ZIKV subgenomic replicon[48]. The replicon, under the control of an SP6 promoter, contained a coding sequence for ZIKV capsid with its anchor domain, prME was replaced with an NLuc reporter, while retaining the last 30 codons of E, followed by non-structural proteins. Using MEGAscript SP6 Transcription kit (Thermo Fisher Scientific), the replicon was in vitro transcribed. To knockdown interactors, AF5 cells seeded at $2.5 \times 10^5$ cells/ well in 24-well plates were transfected with 300 ng of dsRNA or dseGFP as control using 2 μL DharmaFECT 2 in Opti-MEM. After 24 h, cells were co-transfected with 1 μg in vitro transcribed ZIKV Replicon RNA and 50 ng FFLuc expression plasmid (pIZ-Fluc; transfection control) using 2 μL DharmaFECT 2 in Opti-MEM. Cells were lysed 24 hpt using 1× PLB and NLuc and FFLuc levels were measured using a Nano-Glo Dual-Luciferase Reporter Assay System (Promega, Cat# N1610) following the manufacturer's protocol.

**ZIKV genome detection assay.** For the ZIKV genome detection assay at early stages if infection, AF5 cells seeded at $2.5 \times 10^5$ cells/well in 24-well plates were transfected with 300 ng of dsTER94, dsRpn1 or dseGFP control with 2 μL DharmaFECT 2 in Opti-MEM. After 24 hpt, cells were infected with ZIKV PE243 (MOI = 10) on ice with cycloheximide (1 μg/μL) for 30 min to allow simultaneous entry. Any unadhered virus was removed by replacing with fresh L-15 medium with cycloheximide. Cells were then incubated at 28 °C for 1 and 2 h. Total RNA was extracted from each timepoint using TRIzol (Thermo Fisher Scientific) following the manufacturer's instructions and synthesized cDNA used for RT-qPCR of ZIKV PE243 and dsRNA knockdown efficiency.

**siRNA knockdown and infection in A549 cells.** A549 cells were seeded at $1.5 \times 10^5$ cells/well in 24-well plates and transfected with 5 pmol of each or both siRNA against VCP (Thermo Fisher Scientific, Cat# s14767 and s14765), UBR5 (Thermo Fisher Scientific, Cat# s28025 and s224201), SQSTM1 (Thermo Fisher Scientific, Cat# s16960 and s16962), or 5 pmol (siNeg) or 10 pmol (siNeg2X) Silencer Select Negative Control #2 (Thermo Fisher Scientific, Cat# 4390846) as a control with 2 μL DharmaFECT 2 (Horizon Discovery) for 72 h. Cells were infected with ZIKV-NLuc (MOI = 10) or ZIKV PE243 (MOI = 10) for 24 h. Cells were lysed with 1×

PLB for ZIKV-NLuc infected cells or TRIzol for ZIKV PE243 infected cells. Corresponding knockdown efficiencies of each siRNA were checked by immunoblotting using antibodies against VCP, SQSTM1, and UBR5. Cell viabilities 72 hpt were also checked using CellTiter-Glo assay (Promega, Cat# G7571) as per manufacturer's guidelines.

**Chemical inhibitors and infection.** A549 cells seeded at $1.5 \times 10^5$ cells/well in 24-well plates were treated with chemical inhibitors against different proteins in the UPP. Cell viability assays were performed to test the cytotoxicity of the drugs at different concentrations at 24 h post-treatment with CellTiter-Glo assay (Promega, Cat# G7571). A549 cells were treated with Pyr-41 (75 μM), DBeQ (1 μM), EerI (1 μM), ML364 (2 μM), MG132 (20 μM), and DMSO as control (0.1% in DMEM) for 2 h prior to ZIKV PE243 (MOI = 10) infection. Cells were incubated with the drug (or DMSO control) and virus at 37 °C with 5% $CO_2$ for 24 h. Total RNA was then extracted from the cells using TRIzol following the manufacturer's protocol and viral RNA measured by RT-qPCR.

**Drug time-of-addition assay.** For the time-of-addition assay of UPP chemical inhibitors, A549 cells at $1.5 \times 10^5$ cells/well were seeded in 24-well plates and subjected into three different treatments: (1) pre-treatment (pre), (2) co-treatment (co), and (3) post-treatment (post). The same drug concentrations were used as mentioned above. All ZIKV-NLuc (MOI = 10) infections were done in the presence of the inhibitors or DMSO control. In case of pre-treatment, cells were treated with chemical inhibitors or DMSO control for 2 h before infection until 2 hpi. Co-treatment involved incubating the virus with the drug for 1 h at 37 °C before adding the drug with virus on to the cells for 2 h. For the post-treatment, cells were first infected with virus for 2 h and the drugs added until 6 hpi. After each treatment, the drug and inoculum were replaced with fresh media and the cells incubated at 37 °C with 5% $CO_2$ for 24 h. Cells were then lysed with 1× PLB and NLuc readings measured using a GloMax Luminometer following the Nano-Glo Luciferase assay (Promega, Cat# N1130) protocol.

**ZIKV-NLuc infection time course.** A549 cells were seeded at $1.5 \times 10^4$ cells/well in 96-well plates and treated with DBeQ (5 μM), EerI (5 μM), or cycloheximide (1 μg/ μL) for 2 h prior to infection with ZIKV-NLuc (MOI = 10). Infection was performed in the presence of the drugs for 4 to 24 h. Cells were then lysed at 4 h intervals using 1× PLB and NLuc readings measured using Nano-Glo Luciferase Assay system (Promega, Cat# N1130) on GloMax Luminometer (Promega).

**RNAi sensor assay.** As developed previously[66], the RNAi sensor assay AF5-V5-C, AF5-V5-AC, and AF5-V5-eGFP (as control) cells were seeded at $2.5 \times 10^5$ cells/ well in 24-well plates and allowed to settle overnight. Cells were then co-transfected with FFLuc-expressing pIZ-Fluc and *Renilla* luciferase-expressing transfection control (pAcIE1-Rluc) plasmids together with 1 ng dsRNA against FFLuc (dsFluc) or dsLacZ as control for dsRNA cleavage assay or 1 ng siRNA, siFluc or siHyg for siRNA loading assay using 2 μL DharmaFECT 2 in Opti-MEM. After 24 hpt, cells were lysed with 1× PLB and FFLuc/Rluc levels measured using a Dual-Luciferase Reporter Assay System (Promega, Cat# E1969) following the manufacturer's protocol in a GloMax Luminometer (Promega).

**Plaque assay.** For virus titration, cell monolayers of A549-NPro cells at $4 \times 10^5$ cells/well in 12-well plates were infected with serially diluted (1:10) supernatant from TER94, Rpn1, and AeUBR5 knockdown experiments with ZIKV PE243. Dilutions were in DMEM supplemented with 2% FBS and incubated with an overlay consisting of MEM with 4% FBS, 4% HEPES, and 1.2% Avicel under the same conditions as mentioned before for 6 days. Infected cells were then fixed with 4% formaldehyde and stained with 0.2% toluidine blue to visualize plaques.

**In silico analyses.** In generating the PPI networks, only proteins identified in all three replicates of V5-C and V5-AC samples but not in V5-eGFP control samples were considered. UniProt IDs from the protein list were converted to stable gene IDs in VectorBase[52] (www.vectorbase.org) release 49 *beta* using the AaegL5.3 release and used to perform a multiple protein search in StringDB[53] v.11.0 (www. string-db.org) with a minimum required interaction score of 0.7 for each protein. Networks were then exported to Cytoscape v.3.8.2[54] with IP proteomics interactions in gray and StringDB PPI edges in cyan. Interactor nodes are labeled with VectorBase IDs with the mean number of peptides identified and mean LFQ intensity indicated by the size and color, respectively.

All protein interactors used in constructing the PPI network were subjected to GO and pathway enrichment analysis. This was done by performing a database search for gene function in VectorBase and DAVID[55] v.6.8 (https://david.ncifcrf. gov/) using the VectorBase IDs of each protein. GO terms and KEGG IDs with a p-value of <0.05 level of significance with Benjamini correction were considered for the analysis.

To determine *D. melanogaster* orthologs of interactors, OrthoDB[56] v.10.1 (www.orthodb.org) was used. From *D. melanogaster* orthologs, human orthologs were identified from FlyBase[57] FB2020_05 release and confirmed in HGNC[58] (www.genenames.org). If no ortholog was identified, a homology-based approach

was employed using BLASTP limited to the *D. melanogaster* (taxid: 7227) and *H. sapiens* (taxid: 9606) on NCBI (https://blast.ncbi.nlm.nih.gov/Blast.cgi). Only proteins with a high confidence score and similarity were considered. A corresponding PPI network of selected drosophila or human orthologs was constructed in StringDB and visualized in Cytoscape.

Data mining of VCP co-factors was performed in BioGRID[59] v.4.2 (www.thebiogrid.org). A total of 953 unique interactors of VCP was identified. This dataset was compared to the human orthologs of our mosquito interactors data. The overlap between the datasets was 32 proteins. GO analysis in DAVID and VectorBase was performed in this subset of proteins across orthologs. Proteins involved in ubiquitination were assessed by constructing ortholog mapping and PPI networks. Two candidate co-factors, SQSTM1 and UBR5 were further characterized in mosquito and A549 cells.

**Data and statistical analyses.** For ZIKV-NLuc infections, all NLuc readings are presented as relative light units to corresponding NLuc readings of controls set to 1. For dual-luciferase readings, luciferase readings were set as ratios to the luciferase readings of its transfection control and presented as relative light units to ratio luciferase readings of its controls set to 1. Densitometry analysis of bands detected from cycloheximide chase assay immunoblots are presented as a relative signal of the protein of interest to the signal of its corresponding housekeeping protein and the ratio presented relative to the 0 h timepoint set to 1. For siRNA knockdown efficiency immunoblots, densitometry analyses of bands are presented as a relative signal of the target protein to the signal of its corresponding housekeeping protein and the ratio shown relative to the siRNA negative control set to 1. For RT-qPCR results, CT values were transformed into fold change values using the $2^{-\Delta\Delta CT}$ method[118] with S7 and GAPDH internal control genes for mosquito and A549 cells, respectively with error propagation for SDs performed across CT values. Data were obtained from triplicate independent biological experiments, unless specified, with at least three technical repeats per experiment and are presented as mean ± SEM with statistical significance computed using a two-tailed Student's *t*-test or two-way ANOVA when appropriate with the level of significance set at 0.05 or 0.01 with means compared to controls or within specific timepoints to controls. All statistical analyses and data visualizations were done in GraphPad Prism v.7. All values, exact *p*-values, and uncropped immunoblots are provided in the Source Data file.

**Reporting summary.** Further information on research design is available in the Nature Research Reporting Summary linked to this article.

## Data availability

Mass spectrometry RAW files have been deposited to the ProteomeXchange Consortium via the PRIDE partner repository with the dataset identifier PXD020565. Supplemental and experimental data can be accessed through the University of Glasgow Enlighten (https://doi.org/10.5525/gla.researchdata.1020). ZIKV PE243 sequence available from NCBI GenBank (ZIKV/H.sapiens/Brazil/PE243/2015; GenBank: KX197192.1). *Ae. aegypti* mRNA sequences from NCBI RefSeq AaegL5.0 (https://www.ncbi.nlm.nih.gov/assembly/GCF_002204515.2). Publicly available data used in particular analyses were obtained from UniProt[117] (*Ae. aegypti* proteome UP000008820; www.uniprot.org), VectorBase[52] (AaegL5.3; www.vectorbase.org), StringDB[53] (v11.0; www.string-db.org), DAVID[55] (v6.8; https://david.ncifcrf.gov/), OrthoDB[56] (v10.1; www.orthodb.org), FlyBase[57] (FB2020_05; www.flybase.org), HGNC[58] (www.genenames.org), and BioGRID[59] (v4.2; https://thebiogrid.org/113258/summary/homo-sapiens/vcp.html). Further information and reagent requests including generated unique cell lines and plasmids should be addressed to Alain Kohl, alain.kohl@glasgow.ac.uk. Source data are provided with this paper as Source Data file.

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

## Acknowledgements

This work was funded by the UK Medical Research Council (MC_UU_12014/8) for A.K. E.C.H. is funded by an MRC Career Development Award (MR/N008618/1). R.J.G. was supported by a British Council Newton Fund grant, ID 279705176, under the DOST-Newton PhD Scholarship partnership. The grant is funded by the UK Department for Business, Energy and Industrial Strategy, Philippines Department of Science and Technology-Science Education Institute, and the University of the Philippines Visayas and delivered by the British Council. For further information, please visit www.newtonfund.ac.uk.

## Author contributions

Conceptualization, R.J.G., M.V., A.K.; Validation, R.J.G.; Formal analysis, R.J.G., M.V., A.K.; Investigation, R.J.G., J.R., M.V.; Resources, M.V., J.R., C.L.D., D.J.L., A.M., E.C.H.; Data curation, R.J.G., A.K.; Writing—Original draft, R.J.G., M.V., A.K.; Writing—Review and editing, R.J.G., J.R., C.L., D.J.L., E.C.H., A.M., A.K., M.V.; Visualization, R.J.G.; Supervision, M.V., A.K.; Project administration, M.V., A.K.; Funding acquisition, R.J.G., E.C.H., A.K.

## Competing interests

The authors declare no competing interests. The funders had no role in study design, data collection and analysis, decision to publish, or preparation of the manuscript.
