## [Peer Review File · Nature Communications]

REVIEWER COMMENTS

Reviewer #1 (Remarks to the Author):

The authors perform immunoprecipitation and proteomics on Zika virus capsid in *Aedes aegypti* cells to identify mosquito proteins involved in Zika virus replication. They identify TER94/VCP as pro-viral factors in mosquito and human cells, which are likely responsible for degrading capsid upon entry for genome release. The authors also nicely interface established interactions from BioGRID and their new dataset from mosquito cells to uncover UBR5 as a potential TER94/VCP co-factor important for Zika virus replication and capsid degradation. The manuscript is well-written. The virology and cell-based experiments were performed very nicely. The knockdown screen was well done, and the validation of the capsid-TER94 was good to see. The AF5 cell line was an elegant choice (though it may be worth highlighting somewhere in the text why this clone was chosen over the parental Aag2 cell line). The figures are beautifully made and easy to interpret. However, there are several points where this manuscript could be improved:

1. It is unlikely that eGFP captures all of the background associated with IP/proteomics. Figure 1c is therefore an over-representation of capsid-mosquito protein interactions. The discussion of proteomic background was superficial, and a non-expert may interpret these interactions in Figure 1c as high-confidence without understanding their caveats. A deeper discussion of background and a more serious effort to remove background is essential. While most background removal tools focus on mammalian/human systems, more could be done to remove background. The quantitative proteomic data was not even taken advantage of for removing background. Simple scoring systems like SAINT could work well in leveraging a small number of samples and would not require additional data collection.
2. There is an existing abundance of proteomic data that was not leveraged in this study. The authors did not compare their results to unique but related datasets, such as the IP/proteomics studies on Zika virus capsid-human interactions (ref 33-35) and dengue virus capsid-mosquito interactions (ref 34). Was there significant overlap with these datasets?
3. The same could be said of the abundance of existing knockdown/knockout data for flaviviruses. What was the overlap of the proteomic dataset with the Zika virus screens in human cells, or other flavivirus screens in insect cells? Were the pro-viral factors identified in the knockdown screen identified in any of these genome-wide screens?
4. There is no evidence that TER94/VCP and UBR5 are interacting in a complex with Zika virus capsid, or during Zika virus infection. UBR5 could be acting distinctly through another complex. This should be made more clear in the text (results subheading line 302; discussion lines 429-434).

Reviewer #2 (Remarks to the Author):

The authors addressed the role of ZIKV capsid using cutting-edge cell biology technology and conclude on the role of Ubiquitin Proteasome Pathway (UPP) in mediating the release of the virus genome post fusion.

While the cell biology technology is clearly innovative, the main message is not well supported and not so innovative. We knew about the pro-viral role of UPP in flavivirus infections.

The authors based their conclusion on:

1. C in vitro interaction with TER94 and UBR5 using capsid-expressing stable mosquito cells.
2. Pro viral function of TER94 on a ZIKV replicon and virus in mosquito cells at 72hpi
3. TER94 mediates C degradation, which degradation correlated with a decrease of virus genome in mosquito cells
4. Anti viral function of TER94 on a ZIKV replicon in mosquito cells at an unknown time... (the time should be detailed in the legend).
5. TER94/VCP pro viral function is conserved in human cells
6. ubiquitination and proteasome but not deubiquitination post TER94 binding to its target are pro viral.
7. Proteasome is also important for post entry steps
8. identification of UBR5 as a co factor for TER94

All these findings do not support the conclusion. There are unclear points the conclusion.

The role of UPP in ZIKV infection is well supported but was already identified for several other flaviviruses.

Support for a role in capsid release of the genome is mostly based on the use of replicon, which to my knowledge is not appropriate to test this step of the virus cycle. Other information related to this claim is the increase in virus genome upon inhibition of capsid degradation. It is unclear how this decrease in genome degradation is associated to capsid. Estimating translation would be more in line with a pro viral mechanism.

If I got all the information right, there is a discrepancy in the results from the same experiments in two different figures. In fig1b, the authors describe an increase in replicon upon TER94 depletion. However, in fig2f, they describe the opposite and conclude that based on the replicon experiments TER94 has no impact on replication.

Authors should also increase the number of repeats. In the text, they conduct 3 repeat wells most probably from the same plate. The standard in the field is at least 6 well repeats from two different plates, meaning two different cell batches.

Minor changes:

Replicon should be described at its first use and not in fig3.

Error bars for controls in all figures. Even when normalized to 1, there must be error bars.

The abstract is not very clear. The message should be stated more clearly.

We thank the Reviewers for their comments and suggestions. We have reevaluated the proteomics data and conducted more experiments to answer the questions that were brought up. All the responses given in RED.

REVIEWER COMMENTS

Reviewer #1 (Remarks to the Author):

The authors perform immunoprecipitation and proteomics on Zika virus capsid in *Aedes aegypti* cells to identify mosquito proteins involved in Zika virus replication. They identify TER94/VCP as pro-viral factors in mosquito and human cells, which are likely responsible for degrading capsid upon entry for genome release. The authors also nicely interface established interactions from BioGRID and their new dataset from mosquito cells to uncover UBR5 as a potential TER94/VCP co-factor important for Zika virus replication and capsid degradation. The manuscript is well-written. The virology and cell-based experiments were performed very nicely. The knockdown screen was well done, and the validation of the capsid-TER94 was good to see. The AF5 cell line was an elegant choice (though it may be worth highlighting somewhere in the text why this clone was chosen over the parental Aag2 cell line). The figures are beautifully made and easy to interpret.

We thank the reviewer for the encouraging comments. The AF5 cell line was chosen over Aag2 since it has direct Dcr2 knockout version available, designated as AF319 cells. It would permit to assess, whether an identified interacting vector protein is antiviral due to RNAi or not. Though, not that relevant in this case for ZIKV C-TER94 interaction. In addition, future investigations building upon this work could be performed in AF5 cells using CRISPR-Cas9 techniques to generate knockout clones, which we have performed previously (Varjak et al., 2017, Reference 66. This is now mentioned in the text (Lines 482-487).

However, there are several points where this manuscript could be improved:

1. It is unlikely that eGFP captures all of the background associated with IP/proteomics. Figure 1c is therefore an over-representation of capsid-mosquito protein interactions. The discussion of proteomic background was superficial, and a non-expert may interpret these interactions in Figure 1c as high-confidence without understanding their caveats. A deeper discussion of background and a more serious effort to remove background is essential. While most background removal tools focus on mammalian/human systems, more could be done to remove background. The quantitative proteomic data was not even taken advantage of for removing background. Simple scoring systems like SAINT could work well in leveraging a small number of samples and would not require additional data collection.

We have conducted additional analyses of proteomics data using two different scoring systems: a) SAINTq method and b) iBAQ normalization (Lines 101-105) upon consultation with a virus-host interaction expert, Dr Edward C. Hutchinson, who assisted us further on the matter and who is now added to the list of authors on the manuscript. Our initial assessment of potential interactors revealed to be more stringent than these two other approaches and none of the proteins were removed from the list. However, we did conduct a new search against the UniProt database and it is reflected in the manuscript. We do provide the results of two scoring systems in the Supplementary Fig. 1c and as Supplementary Data, which should help readers in the future to compare their results with ours.

2. There is an existing abundance of proteomic data that was not leveraged in this study. The authors did not compare their results to unique but related datasets, such as the IP/proteomics studies on Zika virus capsid-human interactions (ref 33-35) and dengue virus capsid-mosquito interactions (ref 34). Was there significant overlap with these datasets?

Not many protein-protein interactions studies have been conducted in mosquito cells in case of flavivirus infection, or arboviruses in general. Regardless, we compared our proteomics data using the human orthologs with previously published datasets for ZIKV capsid interactome in human cells, and the results are shown a Venn diagram in the Supplementary Fig. 1e (left panel) and mentioned in the text (Lines 158-161). The overlapping set of proteins between different studies was rather small and seems to depend on the cell lines and techniques used. It was of importance to us to verify the interaction between TER94 and ZIKV C via co-immunoprecipitation.

3. The same could be said of the abundance of existing knockdown/knockout data for flaviviruses. What was the overlap of the proteomic dataset with the Zika virus screens in human cells, or other flavivirus screens in insect cells? Were the pro-viral factors identified in the knockdown screen identified in any of these genome-wide screens?

We have also compared our data with previously published high throughput CRISPR screens performed in human cells and provide the relevant information in the Supplementary Fig. 1e (right panel) and in the text (Lines 161-165). Similar to the ZIKV-human PPI studies, minimal overlap between the data sets was observed.

4. There is no evidence that TER94/VCP and UBR5 are interacting in a complex with Zika virus capsid, or during Zika virus infection. UBR5 could be acting distinctly through another complex. This should be made more clear in the text (results subheading line 302; discussion lines 429-434).

An updated data set from BioGrid on VCP interactors was obtained and revealed a more enriched pool of proteins. Conducting the search for TER94/VCP co-factors limited to ubiquitination related proteins across orthologs (Fig. 5b) allowed a more distinct protein interaction network across species (Fig. 5c), highlighting further the novelty of UBR5 as a VCP co-factor during ZIKV infection. In addition, we further tested the interaction between ZIKV C, UBR5, and TER94. Silencing of UBR5 in mosquito cells resulted in the loss of interaction between C and TER94 (Fig. 5f). Furthermore, silencing of UBR5 does not affect ZIKV replicon, similarly to knockdown of TER94 (Fig. 5h). We hope that the reviewer finds these additional data to be significant.

Reviewer #2 (Remarks to the Author):

The authors addressed the role of ZIKV capsid using cutting-edge cell biology technology and conclude on the role of Ubiquitin Proteasome Pathway (UPP) in mediating the release of the virus genome post fusion.

While the cell biology technology is clearly innovative, the main message is not well supported and not so innovative. We knew about the pro-viral role of UPP in flavivirus infections.

The authors based their conclusion on:

1. C in vitro interaction with TER94 and UBR5 using capsid-expressing stable mosquito cells.
2. Pro viral function of TER94 on a ZIKV replicon and virus in mosquito cells at 72hpi
3. TER94 mediates C degradation, which degradation correlated with a decrease of virus genome in mosquito cells
4. Anti viral function of TER94 on a ZIKV replicon in mosquito cells at an unknown time... (the time should be detailed in the legend).
5. TER94/VCP pro viral function is conserved in human cells
6. ubiquitination and proteasome but not deubiquitination post TER94 binding to its target are pro viral.
7. Proteasome is also important for post entry steps
8. identification of UBR5 as a co factor for TER94

All these findings do not support the conclusion. There are unclear points the conclusion. The role of UPP in ZIKV infection is well supported but was already identified for several other flaviviruses.

We thank the reviewer for summarizing the study into clear and concise points. Indeed, there has been more research on flavivirus entry into cells. Unlike these previous studies, we have identified host factors directly involved in the process in human and mosquito cells. We have conducted further experiments, where we show that UBR5 is needed for interaction between ZIKV C and TER94 (Fig 5f, 5h). Thus, we believe that this study is novel and adds significantly to our understanding of mosquito cell-flavivirus interactions.

Support for a role in capsid release of the genome is mostly based on the use of replicon, which to my knowledge is not appropriate to test this step of the virus cycle.

On the contrary, replicons are often used to determine whether virus entry or exit is affected. A replicon-based approach to determine if entry is affected was used in a publication that studies YFV and UPP (<https://www.ncbi.nlm.nih.gov/pmc/articles/PMC7157815/>). Similarly, replicons were used to find and verify novel SARS-CoV-1 (<https://jvi.asm.org/content/87/14/8017>), ZIKV ([https://www.thelancet.com/article/S2352-3964\(17\)30388-2/fulltext](https://www.thelancet.com/article/S2352-3964(17)30388-2/fulltext)), HCV (<https://www.sciencedirect.com/science/article/pii/S0168827814008071>) entry inhibitors, just to give a few examples. We used additional methods as well to determine the matter, including time of addition assays with drugs and a ZIKV infection time course (Supplementary Fig 2e). We could not carry out drug experiment with mosquito cells as the compounds have been developed and characterized for human targets.

Other information related to this claim is the increase in virus genome upon inhibition of capsid degradation. It is unclear how this decrease in genome degradation is associated to capsid. Estimating translation would be more in line with a pro viral mechanism.

Viral RNA that enters the cells is degraded rapidly, inhibition of capsid degradation does not increase viral RNA amount, but rather preserves it (as it is protected). This has been shown for DENV previously by the Gamarnik group (Byk et al., 2016, Reference 75). This matter has been now clarified in the text (Lines 273-285). In addition, we have made additional experiments to show ZIKV replication across a 24 h time course in A549 cells, under VCP inhibited conditions (DBeQ or EerI treated cells) with cycloheximide as a positive control for translation inhibition Supplementary Fig. 2e). The results showed that ZIKV replication was reduced throughout the time course, early on, when VCP was inhibited and the results were comparable to cycloheximide treatment. This is also mentioned in the text (Lines 366-373).

If I got all the information right, there is a discrepancy in the results from the same experiments in two different figures. In fig1b, the authors describe an increase in replicon upon TER94 depletion. However, in fig2f, they describe the opposite and conclude that based on the replicon experiments TER94 has no impact on replication.

We are not exactly sure what figures are being referred in the original manuscript, Fig 1b is on protein interaction network and not on the replicon, and there is no Fig 2f. None of the figures show improved ZIKV replication upon TER94 silencing nor do we claim that TER94 knockdown increases replicon activity. Based on the figure 3f we can see that if virus entry is not inhibited, TER94 knockdown has no statistically significant effect on virus. However, if we use full virus, with functional structural proteins, ZIKV is inhibited, suggesting the involvement of TER94 in virus entry.

Authors should also increase the number of repeats. In the text, they conduct 3 repeat wells most probably from the same plate. The standard in the field is at least 6 well repeats from two different plates, meaning two different cell batches.

We can assure that replicates were done not on the same day but across different weeks, thus, are truly independent. This clarified further in the text (Lines 916-917), and as specified

in the figure legends. All the raw numbers are provided in the Source Data.

Minor changes:

Replicon should be described at its first use and not in fig3.

The replicon system and general design have already been published previously (Mutso et al., 2017, Reference 48) and described there in great details.

Error bars for controls in all figures. Even when normalized to 1, there must be error bars.

We have now redone the figures as per Nature Communications requirements, having individual dots together with bars. All the values are in a spreadsheet and provided as Source Data.

The abstract is not very clear. The message should be stated more clearly.

We have modified the abstract to convey the message better. We hope this is clearer now.

REVIEWER COMMENTS

Reviewer #1 (Remarks to the Author):

The manuscript has been improved considerably. I feel that it is acceptable for publication provided that a few minor revisions are addressed:

1. The proteomic analysis is more rigorous even if it does not change the overall results. One detail is that the Venn diagram in Figure 1b has changed its distribution at some point in the resubmission process, but no explanation is given for why (in the text or the rebuttal). Details regarding the proteomic scoring are not available directly in the manuscript and the "Enlighten" link did not work for me. The change in the Venn diagram needs to be addressed, at least for reviewers. The proteomics scoring details need to be publicly available.
2. The IP data in Figure 5 nicely adds a physical connection between UBR5 and TER95/VCP. No additional information is needed regarding this line of experimentation for publication.
3. In general, the writing is good but could be improved. I encourage the authors to use more concise wording. This effort will improve the clarity of their manuscript and broaden its reach in the long-term.

Reviewer #2 (Remarks to the Author):

The authors made significant efforts to address the issues. However, there is still a mismatch in results between the impact of TER94 in mosquito cells and VCP in human cells when using the replicon. Specifically, VCP kd reduces reporter values (fig 4b), whereas the mosquito homolog TER94 did not (fig 3f). Hence, the homologs do not have the same function or VCP has additional function.

The use of replicon in Fig 4e is thus irrelevant to test the impact of UPP on C with the drug time of addition experiment. This corresponds to the previous point raised about the use of this particular replicon for assessing C degradation function. The hypothesized mechanism is that C degradation through TER94-mediated UPP is required to initiate virus replication. However, the replicon used does not reproduce the uncapsidation of the genome as during virus infection. Nonetheless, it is true that the chemical treatments result in similar results between replicon and viruses. This similarity questions the hypothesized function of TER94.

Although the identification of the C-protein interactors and the role of UPP in virus propagation is of interest, the lack of clear mechanism for TER94 diminishes the interest of the study, which may not meet the quality criteria for Nature communications.

Response to Reviewers, Revision #2

We thank the reviewers for the comments, we have made changes in the manuscript to address their concerns and to provide better clarity regarding the experimental systems that were used in this study. All responses are given in RED.

When revising the paper, we noticed that a legend for Figure 5h had not been added to the text; we apologise for the mistake and have now corrected the issue in this version of the manuscript.

Reviewer #1 (Remarks to the Author):

The manuscript has been improved considerably. I feel that it is acceptable for publication provided that a few minor revisions are addressed:

We thank the reviewer for appreciating the efforts we have made to improve the manuscript.

1. The proteomic analysis is more rigorous even if it does not change the overall results. One detail is that the Venn diagram in Figure 1b has changed its distribution at some point in the resubmission process, but no explanation is given for why (in the text or the rebuttal). Details regarding the proteomic scoring are not available directly in the manuscript and the "Enlighten" link did not work for me. The change in the Venn diagram needs to be addressed, at least for reviewers. The proteomics scoring details need to be publicly available.

The difference in distribution resulted from the re-analysis of the proteomics data as a more appropriate and bespoke *Ae. aegypti* proteome (UP000008820) was downloaded from UniProt, using a version last updated on 29th of June 2020 and it was used as a reference. This proteome set, together with the scoring matrix, is now publicly available with the active Enlighten link, <http://dx.doi.org/10.5525/gla.researchdata.1020>.

We apologise that this link had not been set live at the time of submitting the revision.

2. The IP data in Figure 5 nicely adds a physical connection between UBR5 and TER95/VCP. No additional information is needed regarding this line of experimentation for publication.

We thank the reviewer for this comment.

3. In general, the writing is good but could be improved. I encourage the authors to use more concise wording. This effort will improve the clarity of their manuscript and broaden its reach in the long-term.

We have made an effort to write more concisely and clearer, whilst observing the manuscript guidelines of Nature Communications. One of the co-authors, a native English speaker, of the paper invested additional effort into these changes. The manuscript is now hopefully easier to read.

Reviewer #2 (Remarks to the Author):

The authors made significant efforts to address the issues. However, there is still a mismatch in results between the impact of TER94 in mosquito cells and VCP in human cells when using the replicon. Specifically, VCP kd reduces reporter values (fig 4b), whereas the mosquito homolog TER94 did not (fig 3f). Hence, the homologs do not have the same function or VCP has additional function. The use of replicon in Fig 4e is thus irrelevant to test the impact of UPP on C with the drug time of addition experiment. This corresponds to the previous point raised about the use of this particular replicon for assessing C degradation function. The hypothesized mechanism is that C degradation through TER94-mediated UPP is required to initiate virus replication. However, the replicon used does not reproduce the uncapsidation of the genome as during virus infection. Nonetheless, it is true that the chemical treatments result in similar results between replicon and viruses. This similarity questions the hypothesized function of TER94.

Although the identification of the C-protein interactors and the role of UPP in virus propagation is of interest, the lack of clear mechanism for TER94 diminishes the interest of the study, which may not meet the quality criteria for Nature communications.

We appreciate that the reviewer has noticed our efforts in resolving the comments previously made. We think the reviewer is perhaps not certain where NLuc expressing virus and where replicon have been used, and that has led to some confusion. In Figure 4b, the left panel labelled “ZIKV-NLuc” pertains to the reporter virus, and not the ZIKV NLuc-expressing replicon (abbreviated now to “ZIKV Replicon” throughout (see Figure 3F for example) to keep distinction across the text. The ZIKV-NLuc reporter virus was first described in Line 180-181. ZIKV Replicon was first described in Line 255-257, where we also indicate a difference to ZIKV-NLuc reporter virus; and in the legend of Figure 3 (Line 235-237). Both, ZIKV-NLuc and ZIKV Replicon, are also described in the Methods section. We would like to point out that the ZIKV Replicon was never used in human cells.

Similarly, for Figure 4b, it has been described in the text (Lines 305-306) and figure legend (Lines 316- 318) that virus infection using ZIKV-NLuc reporter virus was performed; compared to Figure 3f, where transfection of *in vitro* transcribed ZIKV Replicon was performed (Line 255-256). Both experimental systems express the NLuc reporter, which may have caused the issue. We hope that careful renaming and explanations above clarify this point. Moreover, Figure 4e and the drug time-of-addition assay does not involve ZIKV Replicon but rather actual infection with ZIKV-NLuc reporter virus, with a specified MOI. This has been mentioned in the text (Lines 352-353) and further explained in the Methods section (Lines 832-834).

Thus, we disagree that there is a “lack of clear mechanism of TER94” as several different experiments have been done to investigate this. The findings in both human and mosquito cells highly complement each other. Specifically, silencing of TER94 in mosquito cells reduces ZIKV replication (Fig 3d); however, knockdown of TER94 does not affect ZIKV based replicon (Fig 3e). These results are in very good accordance with the experiments in human cells: with either NLuc-expressing virus or wild type virus, silencing of VCP (TER94 human ortholog) diminishes virus replication (Fig 4b). This is further strengthened by the time-of-addition assay, where it is found that only in early infection inhibiting VCP affects virus replication (Fig 4d, 4f and Supplementary Fig 2d, 2e). Thus, mosquito and human experiments corroborate each other.

In addition, we would like to stress that the identification of AeUBR5 as a co-factor of TER94 in mosquito cells, which was initially based on interactions of VCP with UBR5 in human cells (from BioGrid), clearly outlines the importance of these interactions to ZIKV infection. This further highlights an interesting similarity of TER94/VCP function during ZIKV infection in two evolutionary distinct species. This experiment was added to the in previous revision of the manuscript, to respond to comments made by Reviewer 1 (Figure 5f).

We hope that the explanations above resolve any issues or misunderstandings over the use of replicon or virus, and interpretation of the data.

REVIEWERS' COMMENTS

Reviewer #1 (Remarks to the Author):

The authors have provided the appropriate information about proteomic scoring and all major concerns have been addressed. However, the Excel tables hosted at Enlighten are not easily parsed. Many other researchers may want to compare these results to other existing studies or future studies. The impact of the work could be raised in the long-term if an easy-to-read supplementary table was included as part of the manuscript that included all of the Uniprot IDs and Gene IDs for the protein interaction network in Figure 1C. I encourage the authors to consider this for this and future manuscripts.

Reviewer #2 (Remarks to the Author):

There was indeed a misunderstanding in the naming of virus reporter and replicon, which both contain nLuc.

After this clarification, the methodological artefact is no longer present.

I support the publication of this study.

Response to Reviewers, Revision #3

We thank the reviewers for the comments, we have made changes in the manuscript to address their concerns and to provide better clarity regarding the experimental systems that were used in this study. All responses are given in RED.

Reviewer #1 (Remarks to the Author):

The authors have provided the appropriate information about proteomic scoring and all major concerns have been addressed. However, the Excel tables hosted at Enlighten are not easily parsed. Many other researchers may want to compare these results to other existing studies or future studies. The impact of the work could be raised in the long-term if an easy-to-read supplementary table was included as part of the manuscript that included all of the Uniprot IDs and Gene IDs for the protein interaction network in Figure 1C. I encourage the authors to consider this for this and future manuscripts.

We thank review for the supportive comment. We actually had the supplementary table included in the previous version of the manuscript and apologise if this was not obvious. We are now referencing the supplementary table in the Figure 1 legend as well.

Reviewer #2 (Remarks to the Author):

There was indeed a misunderstanding in the naming of virus reporter and replicon, which both contain nLuc.

After this clarification, the methodological artefact is no longer present.

I support the publication of this study.

We thank the reviewer for the support.